# DUAL LOTTERY TICKET HYPOTHESIS

**Yue Bai[1], Huan Wang[1], Zhiqiang Tao[2], Kunpeng Li[3], and Yun Fu[1]**
[1]Northeastern University, Boston, MA, USA
[2]Santa Clara University, Santa Clara, CA, USA
[3]Meta Research, Burlingame, CA, USA
{bai.yue, wang.huan}@northeastern.edu
ztao@scu.edu, kunpengli@fb.com, yunfu@ece.neu.edu

## ABSTRACT

Fully exploiting the learning capacity of neural networks requires overparameterized dense networks. On the other side, directly training sparse neural networks typically results in unsatisfactory performance. Lottery Ticket Hypothesis (LTH) provides a novel view to investigate sparse network training and maintain its capacity. Concretely, it claims there exist winning tickets from a randomly initialized network found by iterative magnitude pruning and preserving promising trainability (or we say being in trainable condition). In this work, we regard the winning ticket from LTH as the subnetwork which is in trainable condition and its performance as our benchmark, then go from a complementary direction to articulate the *Dual Lottery Ticket Hypothesis (DLTH): Randomly selected subnetworks from a randomly initialized dense network can be transformed into a trainable condition and achieve admirable performance compared with LTH* — random tickets in a given lottery pool can be transformed into winning tickets. Specifically, by using uniform-randomly selected subnetworks to represent the general cases, we propose a simple sparse network training strategy, Random Sparse Network Transformation (RST), to substantiate our DLTH. Concretely, we introduce a regularization term to borrow learning capacity and realize information extrusion from the weights which will be masked. After finishing the transformation for the randomly selected subnetworks, we conduct the regular finetuning to evaluate the model using fair comparisons with LTH and other strong baselines. Extensive experiments on several public datasets and comparisons with competitive approaches validate our DLTH as well as the effectiveness of the proposed model RST. Our work is expected to pave a way for inspiring new research directions of sparse network training in the future. Our code is available at https://github.com/yueb17/DLTH.

## 1 INTRODUCTION

While over-parameterized networks perform promisingly on challenging machine learning tasks Zagoruyko & Komodakis (2016); Arora et al. (2019); Zhang et al. (2019), they require a high cost of computational and storage resources Wang et al. (2020a); Cheng et al. (2017); Deng et al. (2020). Recent pruning techniques aim to reduce the model size by discarding irrelevant weights of well-trained models based on different criteria Gale et al. (2019); He et al. (2017); Han et al. (2015a;b). Decisive weights are preserved and finetuned for obtaining final compressed model with acceptable performance loss. Following this line, several series of research works have been done to explore effective pruning criterion for better final performances. For example, regularization based pruning approaches Liu et al. (2017); Han et al. (2015a;b) leverage a penalty term during training for network pruning. Also, many researches take advantages of Hessian information to build more proper criteria for pruning LeCun et al. (1990); Hassibi & Stork (1993); Wang et al. (2019a); Singh & Alistarh (2020). However, regular pruning methods still requires a full pretraining with high computational and storage costs. To avoid this, pruning at initialization (PI) attempts determining the sparse network before training and maintains the final performances. For instance, Single-shot Network Pruning (SNIP) Lee et al. (2018) uses a novel criterion called connection sensitivity to measure the weights importance and decide which weights should be removed. Gradient Signal Preservation

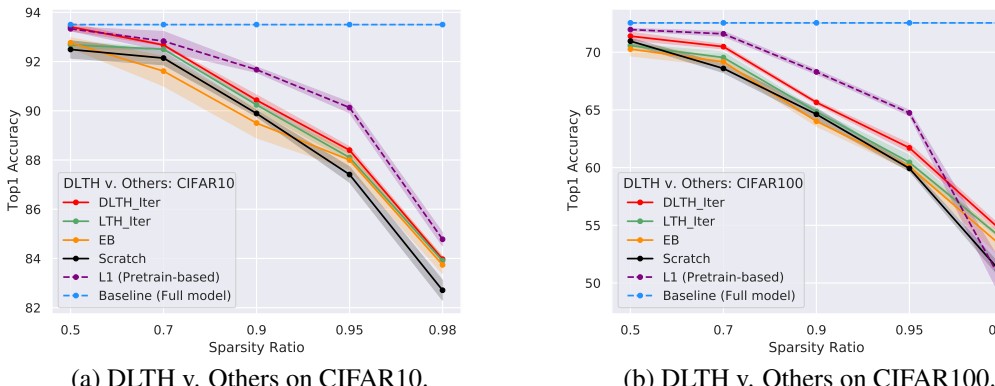

(a) DLTH v. Others on CIFAR10.                    (b) DLTH v. Others on CIFAR100.

Figure 1: Comparisons between DLTH with other training strategies. 3 times average scores with standard deviation shown as shadow are plotted. Both (a)/(b) (CIFAR10/CIFAR100) demonstrate ours superiority in all non-pretrain based approaches (solid lines). Two pretrain based methods (dash lines) are plotted for reference. (This figure is best viewed in color.)

(GraSP) Wang et al. (2020a) regards the gradient flow as an importance factor and correspondingly use it to design the PI criterion.

Regular pruning and PI techniques both achieve promising results on sparse network training with considerable model compression. Nevertheless, they only focus on designating criteria to find a specific sparse network but ignore exploring the internal relationships between the dense network and its eligible subnetwork candidates, which impedes a full understanding of the sparse network training. Lottery Ticket Hypothesis (LTH) Frankle & Carbin (2018) hits this problem with the first shot. It claims a trainable subnetwork exists in the randomly initialized dense network and can be found by pruning a pretrained network. In other words, as long as the dense network is initialized, the good subnetwork has been implicitly decided but needs to be revealed by the pruning process. The LTH has been validated by training the subnetwork from scratch with mask obtained from corresponding iterative magnitude pruning Han et al. (2015b;a). The subnetwork is regarded as the winning ticket among the lottery pool given by the dense network. Accordingly, we treat the subnetworks which can reach the performance of winning ticket from LTH as the ones who are in trainable condition or with promising trainability. This valuable discovery naturally describes the relationship between a random initialized dense network and the trainable subnetwork hidden in it.

However, LTH only focuses on finding one sparse structure at the expense of full pretraining, which is not universal to both practical usage and investigating the relationship between dense and its subnetworks for sparse network training. In our work, we go from a complementary perspective of LTH and propose a new hypothesis called *Dual Lottery Ticket Hypothesis (DLTH)*. It studies a more challenging and general case of investigating the internal relationships between a dense network and its sparse counterparts and achieves promising performances for sparse network training (see Fig. 1). Specifically, DLTH is described as *a randomly selected subnetwork from a randomly initialized dense network can be transformed into a trainable condition and achieve admirable performance using regular finetuning compared with LTH*. In other word, a random ticket in a given lottery pool can be turned into a winning ticket. (Please note, we consider the common cases based on uniformly random selection for weights to obtain subnetworks, not including certain extreme situations such as the disconnected subnetworks.) For convenience, in the following sections, we omit this note and refer to "randomly selected subnetwork" as the common cases mentioned above. To validate our DTLH, we design Random Sparse Network Transformation (RST) to transform a randomly selected subnetwork into a trainable condition. RST simply introduces a gradually increased regularization term to achieve information extrusion from extra weights (which are set to be masked) to the target sparse structure. In this way, we construct a closed-loop research perspective from the dual side of LTH as shown in Fig. 2. Compared with LTH, our DLTH considers a general case – *studying a randomly selected subnetwork instead of a specific one* – with a simple training strategy RST proposed to validate it. The comprehensive relationship between a dense network and its sparse counterparts is inquired for studying sparse network training. Our contributions can be summarized as follows:

- We investigate a novel view of studying sparse network training by presenting *Dual Lottery Ticket Hypothesis (DLTH)*, a dual problem of Lottery Ticket Hypothesis (LTH), articulated as: *A randomly selected subnetwork from a randomly initialized dense network can be transformed into a trainable condition and achieve admirable finetuning performance compared with LTH.*

- We propose a simple sparse network training strategy, Random Sparse Network Transformation (RST), to achieve promising performance and validate our DLTH. Concretely, we realize the information extrusion from the extra weights which are set to be masked to enhance the target sparse structure by leveraging a gradually increased regularization term. In this way, the randomly selected subnetwork will be transformed into a trainable condition.

- Extensive and fair comparisons based on benchmark datasets with competitive approaches using the same finetuning schedule shows our RST obtains promising and consistent performances, solidly demonstrating its effectiveness and validating DLTH.

- Our DLTH, compared with LTH, is more general and challenging to study the relationship between dense network and its sparse counterparts. We expect our work inspires a novel perspective to study sparse neural network in a more flexible and adjustable way.

## 2 RELATED WORK

Network pruning is one of the relevant research topics. In this section, we group the existing pruning methods into *After-Training Pruning* and *Pre-Training Pruning* as follows. In addition, several dynamic sparse network training strategies are proposed to adaptively adjust and train the sparse network referred as *Dynamic Sparse Network Training* below.

**After-Training Pruning.** Most pruning methods prunes a pre-trained network Louizos et al. (2017); Liu et al. (2017); Ye et al. (2018); Han et al. (2015b;a). Algorithms utilize different ranking strategies to pick redundant weights with low importance scores, and remove them to achieve pruning along with acceptable performance drop. Pioneer magnitude-based method Han et al. (2015b;a) regards weights with low values as unnecessary ones, which is straightforward but may remove important low-value weights. Hessian-based approaches measure the weight importance by computing the their removal effects on the final loss LeCun et al. (1990); Hassibi & Stork (1993). Recently published technique also achieves Hessian-free pruning Wang et al. (2020b) by adding regularization which is more computational friendly. In addition, pruning process can be also conducted during the network training. To name a few, a special dropout strategy is utilized in Srinivas & Babu (2016) to adjust the dropout during training and obtain a pruned network after training. A $L_0$ norm regularization based method is proposed for network pruning Louizos et al. (2017). Above algorithms obtaining sparse network from a trained dense network are seen as *After-Training Pruning*.

**Pre-Training Pruning.** This research direction investigates how to prune a randomly initialized network without any training. *Pruning at initialization* is one typical direction deriving sparse network by remove part of randomly initialized weights Lee et al. (2018; 2019); Wang et al. (2020a). For instance, Single-Shot Neural Network Pruning (SNIP) algorithm Lee et al. (2018) first proposes a learnable sparse network strategy according to the computed connection sensitivity. An orthogonal initialization method is proposed to investigate pruning problem in signal propagation view Lee et al. (2019). Gradient Signal Preservation (GraSP) considers to preserve the gradient flow as an efficient criterion to achieve pruning at initialization. On the other hand, Lottery Ticket Hypothesis (LTH) claims that the *winning ticket* subnetwork exists in the randomly initialized network and can be found by deploying conventional pruning algorithm. Then the selected sparse network can be efficiently trained from scratch and achieve promising performance. Based on LTH, Early-Bird (EB) ticket You et al. (2020) proposes an efficient way to find winning ticket. Above algorithms obtaining sparse network from a random dense network are seen as *Pre-Training Pruning*.

**Sparse Network Training.** The sparse network can be dynamically decided and adaptively adjusted through the training to improve the final performance. Sparse Evolutionary Training (SET) Mocanu et al. (2018) proposes an evolutionary strategy to randomly add weights on sparse network and achieve better training. Dynamic Sparse Reparameterization (DSR) Mostafa & Wang (2019) presents a novel direction to dynamically modify the parameter budget between different layers. In this way, the sparse structure can be adaptively adjusted for more effective and efficient usage. Sparse Networks from Scratch (SNFS) Dettmers & Zettlemoyer (2019) designates a momentum

Table 1: Comparisons between DLTH and other settings. `Base model`, `One/Multiple`, `Controllability`, `Transformation`, and `Pretrain` represent where to pick subnetwork, if the setting finds one specific subnetwork or studies randomly selected subnetworks, if sparse structure is controllable, if the selected subnetwork needs transformation before finetuning, and if pretraining dense network is needed, respectively.

| Settings | Base model | One/Multiple | Controllability | Transformation | Pretrain |
|---|---|---|---|---|---|
| Conventional | Pretrained | One | Uncontrollable | No | Yes |
| LTH | Initialized | One | Uncontrollable | No | Yes |
| PI | Initialized | One | Uncontrollable | No | No |
| DLTH (ours) | Initialized | Multiple | Controllable | Yes | No |

based approach as criterion to grow weights and empirically proves it benefits the practical learning performance. Deep Rewiring (DeepR) Bellec et al. (2017) introduces the sparsity during the training process and augments the regular SGD optimization by involving a random walk in the parameter space. It can achieve effectively training on very sparse network relying on theoretical foundations. Rigging the Lottery (RigL) Evci et al. (2020) enhances the sparse network training by editing the network connectivity along with the optimizing the parameter by taking advantages of both weight magnitude and gradient information.

## 3 LOTTERY TICKET PERSPECTIVE OF SPARSE NETWORK

Lottery Ticket Hypothesis (LTH) Frankle & Carbin (2018) articulates the hypothesis: *dense randomly initialized networks contain subnetworks, which can be trained in isolation and deliver performances comparable to the original network*. These subnetworks are seen as *winning tickets* with good trainability and are discovered by iterative magnitude pruning, which requires training a dense network. The mask of the pruned network illustrates the sparse structure of winning tickets.

**Problem Formulation.** We start from introducing general network pruning problem. The neural network training can be seen as a sequence of parameter updating status using stochastic gradient descent (SGD) Wang et al. (2021); Bottou (2010):

$$\{w^{(0)}, w^{(1)}, w^{(2)}, \cdots, w^{(k)}, \cdots, w^{(K)}\}, \tag{1}$$

where $w$ is the model parameter with superscript $k$ as training iteration number. For a general case, the sparse network structure can be mathematically described by a binary mask $m$ which has the same tensor shape as $w$. The process of obtaining $m$ can be formulated as a function $m = F_m(w; D)$, where $D$ is the training data. Further, the weights of the sparse structure are different based on different strategies Hassibi & Stork (1993); Wang et al. (2020b), which is given by $w^* = F_w(w; D)$. The final pruned network can be integrated as

$$\widetilde{w} = F_m(w^{(k_m)}; D) \cdot F_w(w^{(k_w)}; D), \tag{2}$$

where $w^{k_m}$ and $w^{k_w}$ represent different parameter conditions for $F_m$ and $F_w$. The conventional pruning requires that $k_m = k_w = K$. The LTH needs $k_m = K, k_w = 0$, and $F_w = I$, where $I$ is the identical mapping representing model directly inherits the randomly initialized weights.

Against with traditional view that directly training sparse network cannot fully exploit network capacity Wang et al. (2020b;a), LTH validates there exists a sparse network (winning ticket) with better trainability than other subnetworks (other tickets). Specifically, the picked winning ticket will be trained (finetuned) in isolation to evaluate the model performance and illustrate its admirable trainability. However, to uncover this property still needs first pruning a pretrained model to obtain mask and the mask must match with the original dense network - *winning ticket matches with the given lottery pool*. LTH provides a novel angle to understand and reveal the connections between a random dense network and its subnetworks with admirable trainability. However, LTH only validates there exists one specific subnetwork and still requires pruning on pre-trained model to find it, which is a relatively restricted case.

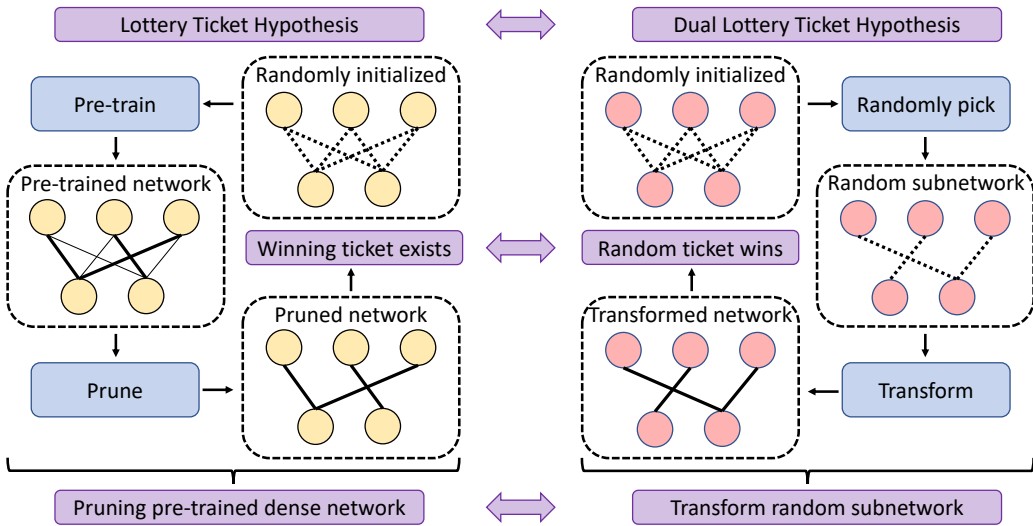

Figure 2: Diagram of Lottery Ticket Hypothesis (LTH) and Dual Lottery Ticket Hypothesis (DLTH).

## 4   DUAL LOTTERY TICKET HYPOTHESIS

We follow the perspective of LTH and investigate sparse network training from a complementary direction. We articulate *Dual Lottery Ticket Hypothesis (DLTH)*. We elaborate our DLTH and discuss its connections and differences with LTH and other related research topics.

> **Dual Lottery Ticket Hypothesis (DLTH).** *A randomly selected subnetwork from a randomly initialized dense network can be transformed into a trainable condition, where the transformed subnetwork can be trained in isolation and achieve better at least comparable performance to LTH and other strong baselines.*

Formally, we follow the general definition provided in Sec. 3 for problem formulation. Our DLTH requires $k_m = 0, k_w = 0$ and allows $F_m$ being random mapping. $F_w$ represents our proposed RST which will be detailed introduced in Sec. 5

**Comparisons with LTH.** LTH proves there *exists* an appropriate subnetwork in a randomly initialized dense network whose weights can be easily trained and overcome the difficulty of training sparse network. The weights of sparse structure are picked from the *initial network* but it requires *pruning pre-trained* dense network to obtain the mask. On the contrary, DLTH claims a *randomly selected* subnetwork can be *transformed* into a proper condition *without pruning* pre-trained model where we can adjustably pre-define the mask and further conduct efficient sparse network training.

On the one hand, these two hypotheses mathematically form a dual statement, which lies in a complementary direction. On the other hand, based on our definition in Sec. 3, a sparse network has two characteristics: 1) network structure; 2) network weights. In this way, LTH can be seen as *finding structure according to weights, because it prunes the pretrained network to find mask using weight magnitude ranking*; DLTH can be seen as *finding weights based on a given structure, because it transforms weights for a randomly selected sparse network.* Therefore, we name our work as DLTH. We clarify the advantages of DLTH compared with LTH from three angles: 1) Although LTH and DLTH are *twins* in mathematics, our DLTH considers a more general scenario (*exist a specific subnetwork versus a random selected subnetwork*) to study the relationship between dense and its sparse networks; 2) DLTH is more valuable for practical usage, because it generally allows to transfer a random subnetwork into a trainable condition instead of pruning a pretrained model to identify one good subnetwork. Further, the transformation is also more efficient than pretraining a dense network in LTH, detailedly discussed in Sec. 6; 3) DLTH allows a flexible sparse network selection but LTH must use the pruned mask and the sparse structure is not controllable.

**Comparisons with Pruning at Initialization (PI).** PI is another relevant research topic, which selects sparse network from scratch according to well-designed criterion. The subnetwork with better trainability is picked for following finetuning. Unlike LTH finding sparse candidate through con-

ventional pruning, PI chooses candidates without training a dense network. Following our definition in Sec. 3, PI requires $k_m = 0, k_w = 0$. $F_m$ is determined by different algorithms and $F_w = I$. Similarly, PI also aims to obtain a specific sparse network based on criteria and the selected sparse structure is not controllable. Our work is not for choosing one specific subnetwork but transferring a random subnetwork into a good condition. In this way, the sparse structure will be flexible and under our control. We summarize these comparisons in Tab. 1.

As LTH and PI use finetuning to evaluate final performance, our DLTH also conducts the same finetuning on the transformed subnetwork to make fair comparison with other methods.

## 5 RANDOM SPARSE NETWORK TRANSFORMATION

Being analogous to the real-world lottery game where the winning ticket exists in the lottery pool but hard to find, randomly picking a subnetwork from a random dense network is not wise and needs to be transformed leveraging on the existing information from other parts of network (other tickets). We follow this analogy to propose Random Sparse Network Transformation (RST) to obtain trainable sparse network. Specifically, we employ a widely-used regularization term Wang et al. (2020b; 2019b) to borrow information from other weights which are set to be masked. It benefits the sparse network training by extruding information from other weights to the target sparse structure.

**Information Extrusion.** We use a regularization term to realize information extrusion. Given a randomly initialized dense network $f(x; \theta)$ with a randomly selected subnetwork represented by mask $m \in \{0, 1\}$, the information extrusion can be achieved as optimizing following loss:

$$\mathcal{L}_R = \mathcal{L}(f(x; \theta), \mathcal{D}) + \frac{1}{2}\lambda \|\theta^*\|_2^2, \tag{3}$$

where loss $\mathcal{L}_R$ contains a regular training loss $\mathcal{L}$ on given data $\mathcal{D}$ and a $L_2$ regularization term added on $\theta^*$. $\theta^*$ is the part of parameter $\theta$ which being masked by $m = 0$. In this way, the unmasked weights, referred as $\overline{\theta^*}$, are regularly trained by SGD, while the masked weights $\theta^*$ are still involved for network training but their magnitudes are gradually suppressed via the increasing regularization. The trade-off parameter $\lambda$ is set to gradually increase from a small value during optimizing $\mathcal{L}_R$:

$$\lambda_{p+1} = \begin{cases} \lambda_p + \eta, & \lambda_p < \lambda_b, \\ \lambda_p, & \lambda_p = \lambda_b, \end{cases} \tag{4}$$

where $\lambda_p$ is the regularization term at $p$-th updating. $\eta$ is the given mini-step for gradually increasing $\lambda$ value. $\lambda_b$ is the bound to limit the $\lambda$ increasing. In this way, the magnitude of $\theta^*$ is diminished from a regular to a small scale along with training process. Hence, the importance of weights is dynamically adjusted from a balanced (equal for all weights) to an imbalanced situation ($\theta^*$ becoming less important with diminishing magnitude). The information of $\theta^*$ is gradually extruded into $\overline{\theta^*}$. Analogically, *the random ticket is transformed to winning ticket by leveraging information from other tickets in the whole lottery pool.* After sufficient extrusion, the $\theta^*$ will have very limited effect on the network and we remove them to obtain the final sparse network, which will be finetuned to obtain the final test performance for model evaluation.

## 6 EXPERIMENTAL VALIDATION

Experiments are based on ResNet56/ResNet18 He et al. (2016) on CIFAR10/CIFAR100 Krizhevsky et al. (2009), and a ImageNet subset Deng et al. (2009) to compare our method with Lottery Ticket Hypothesis (LTH) Frankle & Carbin (2018) and other strong baselines. We set five sparsities: 50%, 70%, 90%, 95%, and 98%. Specifically, we leave the first convolutional layer and the last dense layer as complete, and prune the weights in other middle layers with the consistent sparsity ratio. We run each experiment three times and report their means with standard deviations.

**Comparison Approaches.** We use several comparison methods: L1 Li et al. (2016) is the regular L1 pruning based on pretrained network. LTH Frankle & Carbin (2018) is the lottery ticket performance based on one-shot pruning strategy. LTH-Iter follows the iterative magnitude pruning strategy used in LTH. EB You et al. (2020) is the Early-Bird ticket for LTH with one-shot pruning strategy. Scratch represents training a random subnetwork from scratch without pretraining. RST and RST-Iter are our

Table 2: Performance comparison of ResNet56 on CIFAR10 and CIFAR100 datasets using 50%, 70%, 90%, 95%, and 98% sparsity ratios. Except for the L1 row, the highest/second highest performancs are emphasized with red/blud fonts.

| ResNet56 + CIFAR10: Baseline accuracy: 93.50% | | | | | |
|---|---|---|---|---|---|
| Pruning ratio | 50% | 70% | 90% | 95% | 98% |
| L1 Li et al. (2016) (Pretrain-based) | 93.33±0.08 | 92.83±0.39 | 91.67±0.11 | 90.13±0.21 | 84.78±0.27 |
| LTH Frankle & Carbin (2018) | 92.67±0.25 | 91.88±0.35 | 89.78±0.35 | 88.05±0.50 | 83.85±0.55 |
| LTH Iter-5 Frankle & Carbin (2018) | 92.68±0.39 | 92.50±0.15 | 90.24±0.27 | 88.10±0.36 | 83.91±0.15 |
| EB You et al. (2020) | 92.76±0.21 | 91.61±0.60 | 89.50±0.60 | 88.00±0.38 | 83.74±0.35 |
| Scratch | 92.49±0.35 | 92.14±0.27 | 89.89±0.12 | 87.41±0.31 | 82.71±0.40 |
| RST (**ours**) | 92.34±0.12 | 92.27±0.24 | 90.41±0.05 | 88.24±0.08 | 83.77±0.47 |
| RST Iter-5 (**ours**) | 93.41±0.16 | 92.67±0.02 | 90.43±0.21 | 88.40±0.14 | 83.97±0.09 |

| ResNet56 + CIFAR100: Baseline accuracy: 72.54% | | | | | |
|---|---|---|---|---|---|
| Pruning ratio | 50% | 70% | 90% | 95% | 98% |
| L1 Li et al. (2016) (Pretrain-based) | 71.96±0.10 | 71.59±0.21 | 68.29±0.20 | 64.74±0.22 | 50.04±1.52 |
| LTH Frankle & Carbin (2018) | 69.95±0.47 | 68.24±0.60 | 65.66±0.47 | 60.97±0.30 | 52.77±0.44 |
| LTH Iter-5 Frankle & Carbin (2018) | 70.57±0.15 | 69.54±0.46 | 64.84±0.11 | 60.45±0.61 | 53.83±0.09 |
| EB You et al. (2020) | 70.27±0.59 | 69.16±0.36 | 64.01±0.42 | 60.09±0.33 | 53.14±1.04 |
| Scratch | 70.96±0.25 | 68.59±0.35 | 64.62±0.52 | 59.93±0.24 | 50.80±0.55 |
| RST (**ours**) | 71.13±0.48 | 69.85±0.23 | 66.17±0.18 | 61.66±0.37 | 54.11±0.37 |
| RST Iter-5 (**ours**) | 71.39±0.34 | 70.48±0.19 | 65.65±0.15 | 61.71±0.36 | 54.46±0.32 |

method in one-shot and iterative way, respectively. We use uniform-randomly selected subnetworks to represent the general cases for our RST and validate the DLTH.

**Implementations.** All experiments use the same finetuning schedule for fairness. Experiments on CIFAR10/CIFAR100 are optimized by SGD with 0.9 momentum and 5e-4 weight decay using 128 batch size. Total number of epochs is 200 with 0.1/0.01/0.001 learning rates starting at 0/100/150 epochs, respectively. Those for ImageNet are optimized by SGD with 0.9 momentum and 1e-4 weight decay using 256 batch size. Total number of epochs is 90 with 0.1/0.01/0.001/0.0001 learning rates starting at 0/30/60/75 epochs, respectively.

We use the same schedule as finetuning to pretrain the network for L1 and LTH. For LTH-Iter, we set the total number of epochs as 50 with 0.1/0.01/0.001 learning rates starting at 0/25/37 epochs, respectively, for each mini-pretraining. Other configurations are kept as the same. For EB, we use its original early stop point, 25 epochs (1/8 of total epochs), and keep other settings as the same.

For RST, we set the start point $\lambda_0$ and bound $\lambda_b$ as 0 and 1, respectively. The mini-step $\eta$ is set as 1e-4 and the $\lambda$ is increased every 5 iterations termed as $v_\eta$. Each iteration optimizes the parameter with 64 batch size and 1e-3 learning rate. After $\lambda$ reaches the bound, we continue the extrusion with 40,000 iterations for model stability termed as $v_s$. For RST-Iter, in order to keep fairness with LTH-Iter, we accordingly diminish the transformation process by setting $\lambda$ being increased every 1 iterations and 10,000 iterations for model stability. More detailed comparisons are provided in Sec. 6.1. Since our transformation uses 1e-3 learning rate, we add a small warm-up slot in finetuning for our methods: 10 epochs and 4 epochs with 1e-2 learning rate, for CIFAR10/CIFAR100 and ImageNet, respectively, at the beginning then back to 0.1 as mentioned above.

## 6.1 BASELINE COMPARISONS

Tab. 2 shows the comparisons on CIFAR10/CIFAR100. The full accuracy (first line) is trained by regular SGD on random dense network. For consistency, L1, LTH, LTH Iter-5, and EB inherit the same initial weights as the full accuracy model. Scratch, RST, and RST Iter-5 use the same initial weights as above and they also share the same randomly selected subnetwork from the dense one. Only L1 requires pretrained model and others use random dense network. Note that, once the subnetwork is randomly decided, it is kept intact without any structural modification. Since L1 uses pretrained network, it should have the highest performance. Thus, we exclude the L1 and emphasize the rest rows using red/blue fonts as the highest/second highest performances.

**Performance Analysis.** In Tab. 2, we find: 1) Our method generally outperforms other approaches; 2) Compared with LTH which uses wisely picked subnetwork, our method achieves promising re-

sults on randomly picked subnetwork, which solidly validates our DLTH. Our method also pushes the performance close to the L1 with comparable results and even better in certain cases; 3) Compared with scratch, our method consistently achieves better performance; 4) Iterative strategy benefits both LTH and RST.

**Iterative Cycle Number Choice.** Using iterative magnitude pruning is a key factor for LTH which is the benchmark for our work. Hence, we first conduct an ablation study on LTH to choose the most competitive iteration number (cycle number) for a persuasive comparison. We use CIFAR10 with 0.7 sparsity as an exemplar to set the iterative numbers in Fig. 3. We run 3 times and plot the mean with shadow area as standard deviation. LTH performs better using more iterative cycles; our DLTH takes less advantages of more cycles but still outperforms LTH. To throughly demonstrate our superiority, we set the cycle number as 5 (the best choice for LTH) for both LTH and our DLTH. The ablation study and further analysis of DLTH cycle number is provided in Fig. 4.

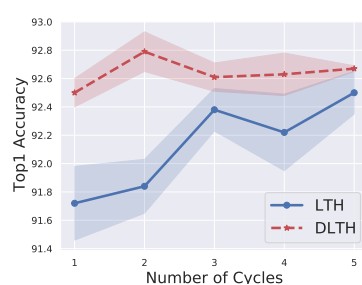

Figure 3: LTH v. DLTH on CIFAR10 with 0.7 sparsity to choose cycle number.

**ImageNet Results.** We compare RST with LTH on a ImageNet subset in Tab. 3. ImageNet subset contains 200 out of 1000 classes from the original dataset. We use the one-shot strategy for LTH/RST and our RST performs better.

Table 3: Test Accuracy of ResNet18 on ImageNet Subset.

| **ResNet18 + ImageNet Subset:** Baseline: 78.96% | | |
|---|---|---|
| Sparsity | 50% | 70% |
| LTH Frankle & Carbin (2018) | 77.87 | 76.48 |
| RST (**ours**) | **78.24** | **76.80** |

**Discussion of DLTH v. LTH.** As a dual format of LTH, DLTH allows randomly selecting subnetwork instead of pruning pretrained network for mask. RST transfers the picked random ticket for winning, instead of directly employing obtained mask to represent the winning ticket in LTH. This pair of hypothesis use the same finetuning schedule and we compare their differences as "pretrain + prune" versus "randomly select + transform": 1) Pretraining requires the same computational resource as finetuning in our experiments. L1 pruning can be done efficiently without any parameter updating; 2) DLTH randomly selects a subnetwork with no extra cost (even no sorting cost). The transformation can be seen as an optimization process. Based on the details in the implementations section above, the extra training cost can be calculated as $c_e = ((\lambda_b/\eta) \cdot v_\eta + v_s) * N_b/N_D$, where $N_b$ and $N_D$ are batch size and number of training samples. Specifically, for RST, this number is (1/1e-4)*5+40000)*64/50000 ≈ 115 epochs (CIFAR as example). We conclude: 1) Since RST still maintains the full network during transformation, the required memory space is the same as pretraining in LTH; 2) Complete LTH and RST require 400 epochs and 315 epochs training cost (both including 200 finetuning epochs). The extra computational cost of RST is lower than LTH. 3) In 5-cycle iterative setting, for RST Iter-5: there are $c_e$ = ((1/1e-4)*1+10000)*64/50000 = 25.6 epochs each cycle and 5 * 25.6 = 128 epochs for total before finetuning; for LTH Iter-5: this number is 5*50 = 250 epochs. RST Iter-5 is still more efficient than LTH Iter-5.

Exactly as saying goes: *There ain't no such thing as a free lunch*, finding trainable sparse structure in a random dense network needs extra cost. As the calculations above, these cost of both LTH and RST are in reasonable scope (ours is more efficient). However, our hypothesis considers a more general and valuable side of this pair-wise hypotheses.

**Iterative Number Ablation Study for RST.** Fig. 3 shows the LTH benefits from the more iterations (cycles) but our RST obtains relatively robust performance with a little improvement. We show the ablations for RST here and leave the detailed discussions in the supplementary material. Fig. 4 visualizes the performance variations of different cycle numbers. We find using iteration strategy (cycles: 2,3,4,5) can outperform the one-shot strategy (cycle: 1) for most cases. However, the overall performances show robustness about iteration number with a little improvement and inapparent correlation between cycle numbers and final performances.

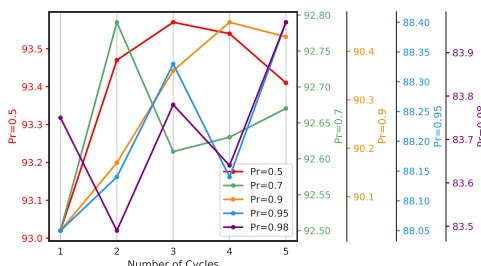 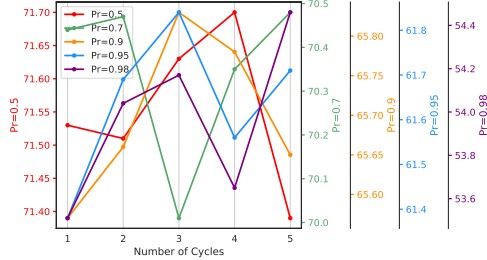

(a) Iterative cycle number ablation on CIFAR10.  (b) Iterative cycle number ablation on CIFAR100.

Figure 4: Ablation study for iterative cycle number of DLTH. (a)/(b) show the 3 times average performance on CIFAR10/CIFAR100 with sparsity ratio from 0.5 to 0.98 and cycle number from 1 to 5. Different colors represent different sparsity ratios. (This figure is best viewed in color.)

**Comparison with GraSP.** We compare RST with Gradient Signal Preservation (GraSP) Wang et al. (2020a), a representative Pruning at Initialization (PI) approach, to further validate our DLTH. Experiments are based on CIFAR10/CIFAR100 datasets (see Tab. 4) using ResNet56 and ImageNet subset (see Tab. 5) using ResNet18 on different sparsities. GraSP selects a subnetwork from a random dense network to finetune. To be fair, RST inherits the weights of the same random dense network and follows the same layer-wise sparsity ratio derived by GraSP. RST achieves better performances using the layer-wise sparsity from GraSP, which means RST has generalibility to handle different sparse structures and further validates our proposed DLTH.

Table 4: Comparison with GraSP on CIFAR10 and CIFAR100 datasets.

| **ResNet56 + CIFAR10:** Baseline accuracy: 93.50% | | | | | |
|---|---|---|---|---|---|
| Sparsity | 50% | 70% | 90% | 95% | 98% |
| GraSP Wang et al. (2020a) | 91.72 | 90.75 | 89.93 | 88.62 | 84.46 |
| RST (**ours**) | **92.69±0.13** | **92.03±0.28** | **90.88±0.11** | **89.14±0.27** | **84.95±0.05** |
| **ResNet56 + CIFAR100:** Baseline accuracy: 72.54% | | | | | |
| Sparsity | 50% | 70% | 90% | 95% | 98% |
| GraSP Wang et al. (2020a) | 67.88 | 67.38 | 64.21 | 59.39 | 45.01 |
| RST (**ours**) | **70.18±0.29** | **69.54±0.07** | **64.86±0.33** | **60.20±0.14** | **45.98±0.36** |

Table 5: Top-1 and top-5 accuracy comparison with GraSP on ImageNet Subset.

| **ResNet18 + Imagenet Subset** | | | | | |
|---|---|---|---|---|---|
| Sparsity | 70% | | 90% | | 95% | |
| Metric | Acc@1 | Acc@5 | Acc@1 | Acc@5 | Acc@1 | Acc@5 |
| GraSP Wang et al. (2020a) | 75.20 | 92.17 | 73.64 | 91.15 | 69.70 | 89.62 |
| RST (**ours**) | **76.95** | **92.87** | **74.49** | **91.52** | **70.33** | **90.01** |

# 7 CONCLUSION AND DISCUSSION

We propose a Dual Lottery Ticket Hypothesis (DLTH) as a dual problem of Lottery Ticket Hypothesis (LTH). We find that a randomly selected subnetwork in a randomly initialized dense network can be transformed into an appropriate condition with admirable trainability. Compared with LTH, our DLTH considers a more challenging and general case about studying sparse network training, being summarized as follows: 1) It is executable to transferring a randomly selected subnetwork of a randomly initialized dense network into a format with admirable trainability; 2) The flexibility of selecting sparse structure ensures us having the controllability of the subnetwork structure instead of determined by pruning method; 3) Our DLTH studies sparse network training in a complementary direction of LTH. It investigates general relationships between dense network and its sparse subnetworks which is expected to inspire following research for sparse network training. We propose a simple strategy, Random Sparse Network Transformation (RST), to validate our DLTH. Specifically, we naturally involve a regularization term and leverage other weights to enhance sparse network learning capacity. Extensive experiments on several datasets with competitive comparison methods substantially validate the DLTH and effectiveness of the proposed RST.

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

## A    ANALYSIS OF ITERATIVE NUMBER ABLATION STUDY FOR DLTH

We analyze the ablation pattern from implementation aspect of iterative strategy for both LTH and RST. Then, we provide more experimental details. For both LTH and RST with iteration strategy, the corresponding subnetwork is selected or pruned iteratively until it reaches the given sparsity. However, the iterative operations for LTH and RST are different. For each iteration of LTH, it obtains the pretrained model and uses L1 pruning to decide the subnetwork based on the weight magnitude. In this way, the subnetwork is elaboratively selected within each iteration and LTH is progressively enhanced to achieve higher performance. On the other side, within each iteration, RST randomly selects weights to add regularization and removes them at the end of this iteration. This iteration strategy does improve the performances for most cases (see comparison between RST and RST Iter-5 in Tab. 2), because the extra weights are gradually removed through each iteration and the whole transformation is more stable. However, compared with LTH enhanced by iteratively using L1 pruning, this progressive enhancement for RST is relatively weak. We further analyze this comparison based on the experiments of ablation study.

Table 6: Cycle number ablation study for RST ResNet56 on CIFAR10 and CIFAR100 datasets.

| ResNet56 | CIFAR10: Baseline accuracy: 93.50% | | | | | CIFAR100: Baseline accuracy: 72.54% | | | | |
|---|---|---|---|---|---|---|---|---|---|---|
| Cycles | 50% | 70% | 90% | 95% | 98% | 50% | 70% | 90% | 95% | 98% |
| 1 | 93.02 | 92.50 | 90.03 | 88.05 | 83.75 | 71.53 | 70.44 | 65.57 | 61.38 | 53.51 |
| 2 | 93.47 | 92.79 | 90.17 | 88.14 | 83.49 | 71.51 | 70.47 | 65.66 | 61.69 | 54.04 |
| 3 | 93.57 | 92.61 | 90.36 | 88.33 | 83.78 | 71.63 | 70.01 | 65.83 | 61.84 | 54.17 |
| 4 | 93.54 | 92.63 | 90.46 | 88.14 | 83.64 | 71.70 | 70.35 | 65.78 | 61.56 | 53.65 |
| 5 | 93.41 | 92.67 | 90.43 | 88.40 | 83.97 | 71.39 | 70.48 | 65.65 | 61.71 | 54.46 |
| Mean | 93.40 | 92.64 | 90.29 | 88.21 | 83.73 | 71.55 | 70.35 | 65.70 | 61.64 | 53.97 |
| Std | 0.20 | 0.09 | 0.16 | 0.13 | 0.16 | 0.11 | 0.18 | 0.09 | 0.16 | 0.35 |

Experimentally, Tab. 6 provides performance means and standard deviations of each cycle number for RST ablation study on CIFAR10/CIFAR100 datasets. We find RST obtains relatively robust performances among different cycle numbers. However, it generally achieves better performances with small standard deviations compared with other methods in Tab. 2. We conclude RST is adaptable and robust for iteration strategy and even obtains better performances using fewer cycles compared with LTH, which can further save the computational costs.

In order to make fair comparisons with LTH, we implement RST with an iteration strategy following the fashion in LTH and use the same iterative number. For how to wisely leverage the iteration strategy for RST to obtain further improvement, we leave it into our future work.

## B    DETAILED EXPERIMENTAL COMPARISON

As an extension of experimental comparisons in the tables of **Baseline Comparisons** section, we provide the performances of each run for L1, LTH, Scratch, and RST with their average and standard deviation shown in Tab. 7 and Tab. 8, which demonstrate the consistent effectiveness of our proposed RST. The column of Mean±Std is the same values in the tables of our main text.

## C    VISUALIZATIONS OF PERFORMANCE COMPARISON

We provide visualizations of performance comparisons. The comparison between Lottery Ticket Hypothesis (LTH) Frankle & Carbin (2018) and our Dual Lottery Ticket Hypothesis (DLTH) using the proposed Random Sparse Network Transformation (RST) on CIFAR10 and CIFAR100 Krizhevsky et al. (2009) are shown in Fig. 5 and Fig. 6. The comparisons between our RST and training from scratch on CIFAR10 and CIFAR100 Krizhevsky et al. (2009) datasets are shown in Fig. 7 and Fig. 8. We plot the 100 to 200 finetuning epochs for a clear visualization.

## D    IMPLEMENTATION DETAILS

We use 4 NVIDIA Titan XP GPUs to perform our experimental evaluations. Each experiment on CIFAR10/CIFAR100 dataset requires around 6 hours on one GPU and each experiment on ImageNet Deng et al. (2009) subset requires around 12 hours on two GPUs. Further, in the section of comparison between our RST and GraSP Wang et al. (2020a) algorithm, we use the layer-wise sparsity ratio obtained from GraSP. We provide the detailed layer-wise ratio for CIFAR10 and CIFAR100 shown in Fig. 9 and Fig. 10, respectively.

## E    CURRENT LIMITATION AND FUTURE WORK

Our Dual Lottery Ticket Hypothesis (DLTH) paves a complementary direction to explore Lottery Ticket Hypothesis (LTH) problem. Current experimental results and analysis substantially validate our proposed DLTH. However, there are still some remaining problems. For example, we validate the random sparse network can be manipulated into a good trainable condition in this work. However, which type of sparse structure can be transformed in a high efficiency and which fits the practical deployment appropriately to benefit computational acceleration? These questions are still open and we will continually explore them in our future work.

Table 7: Different runs details of test accuracy comparisons on CIFAR10 dataset using ResNet56. Our method generally achieves better performance and validates the proposed DLTH.

| ResNet56 + CIFAR10: Baseline accuracy: 93.50% | | | | |
|---|---|---|---|---|
| Pruning ratio 50 % | Run # 1 | Run # 2 | Run # 3 | Mean±Std |
| L1 Li et al. (2016) | 93.40 | 93.22 | 93.36 | 93.33±0.08 |
| LTH Frankle & Carbin (2018) | 92.99 | 92.65 | 92.37 | 92.67±0.25 |
| Scratch | 92.47 | 92.07 | 92.93 | 92.49±0.35 |
| RST (**ours**) | 92.43 | 92.17 | 92.42 | 92.34±0.12 |
| Pruning ratio 70% | Run # 1 | Run # 2 | Run # 3 | Mean±Std |
| L1 Li et al. (2016) | 93.26 | 92.31 | 92.93 | 92.83±0.39 |
| LTH Frankle & Carbin (2018) | 92.34 | 91.49 | 91.82 | 91.88±0.35 |
| Scratch | 92.50 | 92.07 | 91.84 | 92.14±0.27 |
| RST (**ours**) | 92.26 | 91.98 | 92.57 | 92.27±0.24 |
| Pruning ratio 90% | Run # 1 | Run # 2 | Run # 3 | Mean±Std |
| L1 Li et al. (2016) | 91.62 | 91.57 | 91.82 | 91.67±0.11 |
| LTH Frankle & Carbin (2018) | 89.30 | 89.89 | 90.14 | 89.78±0.35 |
| Scratch | 89.97 | 89.73 | 89.98 | 89.89±0.12 |
| RST (**ours**) | 90.39 | 90.35 | 90.48 | 90.41±0.05 |
| Pruning ratio 95% | Run # 1 | Run # 2 | Run # 3 | Mean±Std |
| L1 Li et al. (2016) | 90.19 | 90.35 | 89.85 | 90.13±0.21 |
| LTH Frankle & Carbin (2018) | 88.41 | 87.35 | 88.40 | 88.05±0.50 |
| Scratch | 87.66 | 86.97 | 87.59 | 87.41±0.31 |
| RST (**ours**) | 88.36 | 88.21 | 88.16 | 88.24±0.08 |
| Pruning ratio 98% | Run # 1 | Run # 2 | Run # 3 | Mean±Std |
| L1 Li et al. (2016) | 85.15 | 84.67 | 84.52 | 84.78±0.27 |
| LTH Frankle & Carbin (2018) | 84.11 | 84.35 | 83.08 | 83.85±0.55 |
| Scratch | 82.82 | 82.17 | 83.14 | 82.71±0.40 |
| RST (**ours**) | 83.94 | 83.13 | 84.24 | 83.77±0.47 |

Table 8: Different runs details of test accuracy comparisons on CIFAR100 dataset using ResNet56. Our method generally achieves better performance and validates the proposed DLTH.

| **ResNet56 + CIFAR100:** Baseline accuracy: 72.54% | | | | |
|---|---|---|---|---|
| Pruning ratio 50 % | Run # 1 | Run # 2 | Run # 3 | Mean±Std |
| L1 Li et al. (2016) | 71.95 | 71.85 | 72.09 | 71.96±0.10 |
| LTH Frankle & Carbin (2018) | 70.08 | 69.32 | 70.46 | 69.95±0.47 |
| Scratch | 71.18 | 71.09 | 70.61 | 70.96±0.25 |
| RST (**ours**) | 70.56 | 71.74 | 71.08 | 71.13±0.48 |
| Pruning ratio 70% | Run # 1 | Run # 2 | Run # 3 | Mean±Std |
| L1 Li et al. (2016) | 71.35 | 71.54 | 71.87 | 71.59±0.21 |
| LTH Frankle & Carbin (2018) | 69.08 | 67.95 | 67.70 | 68.24±0.60 |
| Scratch | 68.61 | 69.01 | 68.16 | 68.59±0.35 |
| RST (**ours**) | 69.81 | 69.59 | 70.16 | 69.85±0.23 |
| Pruning ratio 90% | Run # 1 | Run # 2 | Run # 3 | Mean±Std |
| L1 Li et al. (2016) | 68.55 | 68.25 | 68.07 | 68.29±0.20 |
| LTH Frankle & Carbin (2018) | 65.08 | 66.23 | 65.67 | 65.66±0.47 |
| Scratch | 64.33 | 64.18 | 65.34 | 64.62±0.52 |
| RST (**ours**) | 66.42 | 65.98 | 66.11 | 66.17±0.18 |
| Pruning ratio 95% | Run # 1 | Run # 2 | Run # 3 | Mean±Std |
| L1 Li et al. (2016) | 64.57 | 64.61 | 65.05 | 64.74±0.22 |
| LTH Frankle & Carbin (2018) | 60.55 | 61.22 | 61.15 | 60.97±0.30 |
| Scratch | 59.89 | 59.66 | 60.24 | 59.93±0.24 |
| RST (**ours**) | 61.77 | 62.05 | 61.17 | 61.66±0.37 |
| Pruning ratio 98% | Run # 1 | Run # 2 | Run # 3 | Mean±Std |
| L1 Li et al. (2016) | 52.19 | 48.88 | 49.04 | 50.04±1.52 |
| LTH Frankle & Carbin (2018) | 53.38 | 52.53 | 52.39 | 52.77±0.44 |
| Scratch | 51.36 | 51.00 | 50.05 | 50.80±0.55 |
| RST (**ours**) | 53.84 | 53.85 | 54.63 | 54.11±0.37 |

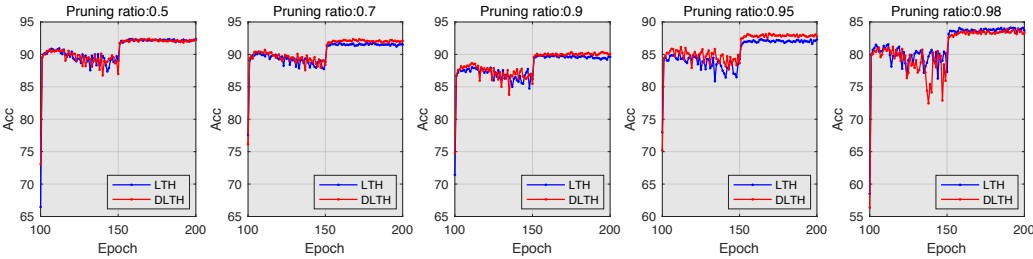

Figure 5: Visualization of test accuracy comparison of LTH and DLTH on CIFAR10 using ResNet56. Our DLTH general outperforms LTH and achieves comparable performance for 98% pruning ratio.

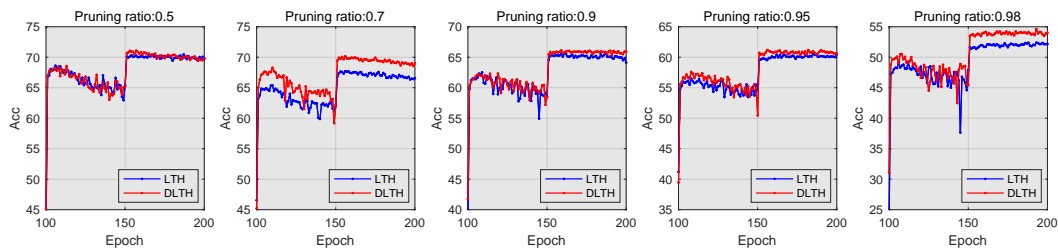

Figure 6: Visualization of test accuracy comparison of LTH and DLTH on CIFAR100 using ResNet56. Our DLTH general outperforms LTH for all sparsities.

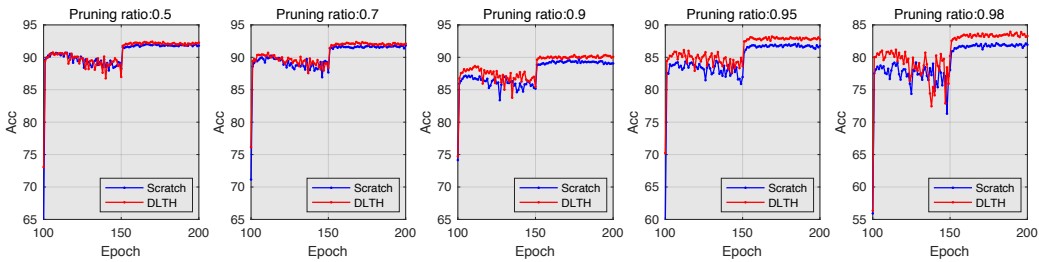

Figure 7: Visualization of test accuracy comparison of training from scratch and DLTH on CIFAR10 using ResNet56. Our training strategy consistently surpasses training from scratch on different pruning ratios. The performance gain increases when the pruning ratio becomes larger.

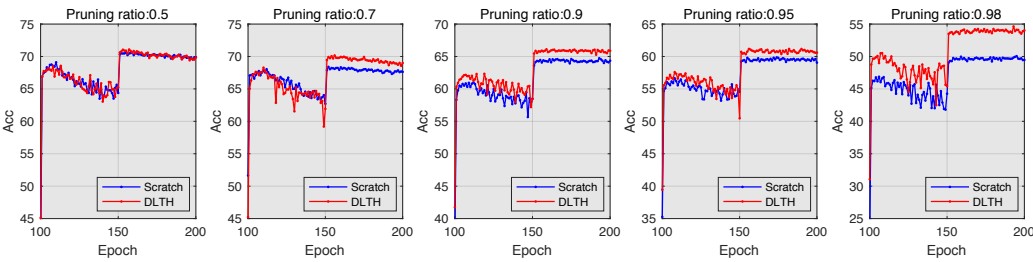

Figure 8: Visualization of test accuracy comparison of training from scratch and DLTH on CIFAR100 using ResNet56. Our training strategy consistently surpasses training from scratch on different pruning ratios. The performance gain increases when the pruning ratio becomes larger.

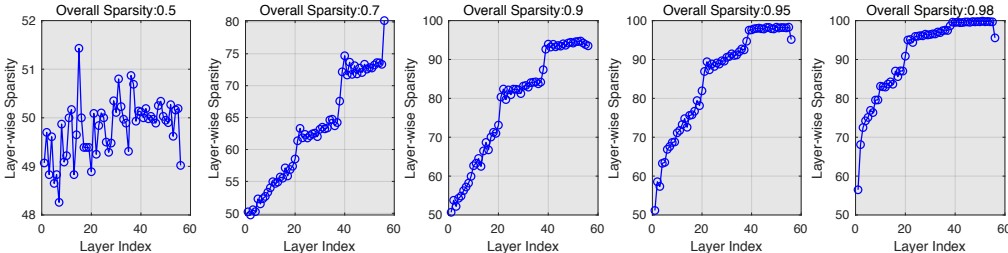

Figure 9: ResNet56 layer-wise sparsity ratio of different overall sparsity on CIFAR10 dataset. These layer-wise ratios are obtained by GraSP Wang et al. (2020a) algorithm. The last layer is the final fully-connected layer whose value is relatively special compared with others. The layer-wise sparsity generally starts from around 50% and increases when layer index increases for different overall sparsity (except for 50% whose layer-wise sparsity are distributed around 50% for all layers).

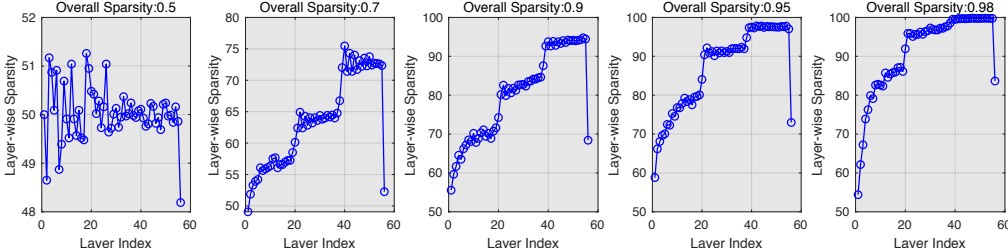

Figure 10: ResNet56 layer-wise sparsity ratio of different overall sparsity on CIFAR100 dataset. These layer-wise ratios are obtained by GraSP Wang et al. (2020a) algorithm. The last layer is the final fully-connected layer whose value is relatively special compared with others. The layer-wise sparsity generally starts from around 50% and increases when layer index increases for different overall sparsity (except for 50% whose layer-wise sparsity are distributed around 50% for all layers).

