# OpenReview forum: "Dual Lottery Ticket Hypothesis"
_ICLR.cc/2022/Conference — ICLR 2022 Poster_

### Official Review · Reviewer_jr1G · 2021-10-20

**Correctness:** 3
**Technical Novelty And Significance:** 3
**Empirical Novelty And Significance:** 3
**Recommendation:** 8
**Confidence:** 4

**Main Review:**

Strength:

1. The concept of the Dual Lottery Ticket Hypothesis is interesting and worth studying.
2. The construction of the paper is clear and easy to follow.

Weakness:

1. As formulated in the abstract, the DLTH demonstrates that any ticket in a given lottery pool can be transformed into a winning ticket. And the definition of winning tickets is a subnetwork, which can be trained from scratch and match the performance of the dense counterpart. But on CIFAR-10/100 and ImageNet with a 50-98% pruning ratio, the performance of all the subnetworks still remains a unignorable gap against the dense model. Personally, I think it is more meaningful to investigate the proposed method in the setting when subnetworks identified by both LTH and DLTH are winning tickets.

2. As suggested in [1], the connections of neurons in individual layers in initial tickets can be totally rearranged without performance drop, and only the layer-wise sparsity level matters. The RST method follows the same layer-wise sparsity ratio derived by GraSP. Is the proposed RST still effective when the layer-wise sparsity level is also randomly arranged?

3. The related work about sparse network training needs to be re-organized, which only tells several works without any logic.

[1] Su et al. Sanity-Checking Pruning Methods: Random Tickets can Win the Jackpot, NeurIPS 2020

Minor:

1. The caption of Figure 3 covers part of the y label.

**Summary Of The Paper:**

This paper extends the Lottery Ticket Hypothesis(LTH) to a more challenging case and proposes the concept of Dual LTH: Any randomly selected subnetwork of a randomly initialized dense network can be transformed into a winning ticket. And the authors propose Random Sparse Network Transformation(RST) to accomplish the transformation process. Experiment results on CIFAR-10/00 and part of ImageNet with ResNet-18/56 demonstrate the effectiveness of the RST and validate the Dual LTH.

**Summary Of The Review:**

Overall, it's interesting to investigate the proposed Dual Lottery Ticket Hypothesis. However current results are a little bit insufficient to validate the DLTH. Specifically, no winning tickets have been identified in current experiments and whether DLTH can be extended to the setting when layer-wise sparsity level can also to randomly arranged.

---

> ### Author Response · Authors · 2021-11-20
> **Response To Reviewer jr1G**
>
> We appreciate the Reviewer's approval and constructive suggestions for us to improve our work. We make the response as below.
>
> **Different pruning ratios to compare with the performance of the dense counterpart:** Based on our current results in Table2 and Table4, we find both LTH and DLTH with pruning ratio 0.5 achieve comparable performance to the dense model with a little bit of performance drop. Besides, we follow the Reviewer's suggestions to further add LTH and DLTH experiments with pruning ratios 0.1 and 0.3 on CIFAR10/CIFAR100. Both of them are based on the iterative strategy with 5 cycles. We also include the results of pruning ratio 0.5 from Table2 for comparisons and the results are shown below.
>
> | cifar10 | pr0.1 | pr0.3 | pr0.5 |
> |:----    | :----:| :----:| :----:|
> | Dense model accuracy: 93.50|
> |LTH Iter-5|93.82 (0.21)|92.66 (0.42)| 92.68 (0.39)|
> |RST Iter-5|**93.86 (0.11)** | **93.62 (0.18)** | **93.41 (0.16)** |
>
> ----------------
>
> | cifar100 | pr0.1 | pr0.3 | pr0.5 |
> |:----    | :----:| :----:| :----:|
> | Dense model accuracy: 72.54|
> |LTH Iter-5|71.63 (0.15)|71.64 (0.39)|70.57 (0.15)|
> |RST Iter-5| **72.74 (0.24)** | **72.40 (0.30)** | **71.39 (0.34)** |
>
> ----------------
>
> All results are the average of three-time runs with standard deviation. We find both LTH and DLTH (referred as RST Iter-5) obtain comparable even better performances to the dense model accuracy for all three pruning ratios on both CIFAR10 and CIFAR100. Our DLTH generally outperforms LTH. In low pruning ratios, DLTH even achieves higher performances than the dense model accuracy. These experimental observations further support our conclusions.
>
> **Layer-wise pruning ratio:** We follow the layer-wise pruning ratio obtained from GraSP to make fair comparisons in our draft. For the effectiveness of RST on other general random layer-wise sparsity settings, actually, we can directly compare the results in Table3 with those in Table2. We put them together to compare as below.
>
> |cifar10|pr0.5|pr0.7|pr0.9|pr0.95|pr0.98|
> |:----|:----|:----|:----|:----|:----|
> |GraSP|91.72 |90.75 |89.93 |88.62 |84.46|
> |RST (GraSP layer-wise sparsity)| **92.69 (0.13)** | 92.03 (0.28) | **90.88 (0.11)** | **89.14 (0.27)** | **84.95 (0.05)** |
> |RST (uniform) |92.34 (0.12) | **92.27 (0.24)** | 90.41 (0.05)| 88.24 (0.08) |83.77 (0.47)|
>
> |cifar100|pr0.5|pr0.7|pr0.9|pr0.95|pr0.98|
> |:----|:----|:----|:----|:----|:----|
> |GraSP|67.88 |67.38 |64.21 |59.39 |45.01 |
> |RST (GraSP layer-wise sparsity)| 70.18 (0.29)| 69.54 (0.07)| 64.86 (0.33)| 60.20 (0.14)| 45.98 (0.36)|
> |RST (uniform) | **71.13 (0.48)** | **69.85 (0.23)** | **66.17 (0.18)** | **61.66 (0.37)** | **54.11 (0.37)** |
>
> RST (uniform) uses the consistent layer-wise sparsity which is a common and general case compared with that from GrsSP. We find the layer-wise sparsity from GraSP helps for better performances on CIFAR10 but cannot make improvements on CIFAR100. Our RST on general layer-wise sparsity achieves promising performances on CIFAR10 and better results on CIFAR100 which shows its effectiveness and generalizability. Both RST (GraSP layer-wise sparsity) and RST (uniform) use a simple one-shot strategy.
>
> Hope our explanation helps clear the Reviewer's concerns.
>
> For the Reviewer's other comments about writing of related work section and figure format, we will further polish our draft to deliver a more decent version.

---

> > ### Comment · Reviewer_jr1G · 2021-11-22
> > **Feedbacks based on the authors' response**
> >
> > Dear authors,
> >
> > Many thanks for all the efforts and extensive new experiments. My concerns are well-addressed and I have raised my score to 8.
> >
> > Best wishes,
> >
> > Reviewer jr1G

---

> > > ### Author Response · Authors · 2021-11-23
> > > **Response To Reviewer jr1G**
> > >
> > > We appreciate the Reviewer's approval for our response and recognition of our work.
> > >
> > > Best wishes,
> > > Paper181 Authors

---

### Official Review · Reviewer_mwor · 2021-10-25

**Correctness:** 3
**Technical Novelty And Significance:** 3
**Empirical Novelty And Significance:** 2
**Recommendation:** 8
**Confidence:** 4

**Main Review:**

The idea of exploring a transferable LTH is interesting, which is complementary to the original LTH. The empirical study is clean and shows promising results on some datasets. In addition, the paper also compares DLTH with other techniques such as Pruning at Initialization (PI). The general writing is good.

However, I have a few concerns about the current version:
1. The empirical study is only on small backbones and datasets, how about larger ones (ImageNet, etc.)?
2. The paper only introduces the regularization method to transform the RST to winning tickets. It is better to have a deep dive into what kind of approaches could work.
3. The paper ignores some related but important works, such as [a,b,c]. It would be better to compare with them, e.g., RigL.
    a. Rigging the lottery: Making all tickets winners.
    b. The Elastic Lottery Ticket Hypothesis.
    c. Drawing early-bird tickets: Toward more efficient training of deep networks.

**Summary Of The Paper:**

This paper studies the Lotter Ticket Hypothesis (LTH) and proposes a Dual Lottery Ticket Hypothesis (DLTH). DLTH describes that any ticket in a given lottery pool can be transformed into a winning ticket. The paper uses a regularization-based method for this transformation. The experiments have been conducted on CIFAR10/100 with ResNets, which indicates a consistent empirical result with DLTH.

**Summary Of The Review:**

Overall, this is an interesting and good paper. If the authors can address my concerns in the rebuttal phase. I will consider making the recommendation for the paper.

-----------------------------------------------------------------------------------------
Most of my concerns have been addressed by the authors. I change my score and recommendation.

---

> ### Author Response · Authors · 2021-11-21
> **Response To Reviewer mwor**
>
> We appreciate the Reviewer's approval and valuable comments. We respond to the Reviewer's concerns as below.
>
> **ImageNet dataset:** We follow the Reviewer's suggestions to make experiments on the ImageNet dataset using ResNet18 with pruning ratios 0.5 and 0.7. The results are shown below.
>
> |ImageNet|pr0.5|pr0.7|
> |----|----|----|
> |Dense model accuracy: 70.39|
> |LTH|67.79|**65.85**|
> |RST|**67.92**|65.54|
> ------------
>
> Both "LTH" and "RST" use the one-shot strategy. We find that compared with dense model accuracy, RST achieves comparable performances to LTH. On this large-scale dataset, our DLTH, as a dual problem of original LTH, obtains promising results compared with its LTH counterpart which also supports our conclusions.
>
> **The method of random network transformation:** We agree with the Reviewer's suggestions about further method exploration of how to conduct the random network transformation and obtain the winning ticket. In our work, we involve a simple yet effective regularization term to achieve the transformation. The RST is naturally proposed to gradually exclude information from other weights into the given sparse structure, which can be analogically described as -- *A randomly picked ticket is manipulated into a winning ticket by leveraging the information from other parts of the given lottery pool*. Based on our current understanding, we think the promising exploration direction is still how to efficiently exploit the pruned weights of the dense model (other parts of the lottery pool) to improve the sparse network performance (randomly picked ticket). We will keep exploring this direction accordingly.
>
> **Related works:** We thank for the Reviewer pointing out these important related works. Concretely, **RigL** [1] proposes a dynamic strategy to update the weights and adjust the sparse network structure simultaneously. However, our DLTH focuses on first fixing a randomly picked sparse network then transforming its weights to obtain a trainable sparse network. Our DLTH actually has a different setting with RigL and we will discuss this difference in our draft; **ELTH** [2] explores the transferability of LTH from source to target network and obtains promising performances. However, the ELTH has comparable but still a little bit lower results compared to the winning ticket found by using iterative magnitude pruning (IMP) on the target network itself, which is named as "LTH Iter-5" in the Table2 of our draft. We have shown the effectiveness of RST and validated DLTH based on the comparisons with "LTH Iter-5" in Table2 and we will supplement this discussion for ELTH in our final version; **Early-bird ticket (EB)** [3] finds the winning ticket in the early pre-training process to achieve high efficiency. This is a highly relevant work to our DLTH. We have already compared with EB in Table2. All these three works are highly valuable and we will include and discuss them in the related work section in our final version.
>
> Hope our response can help for a better understanding of our DLTH and clear the Reviewer's concerns.
>
> [1] "Rigging the Lottery: Making All Tickets Winners", Utku Evci, Trevor Gale, Jacob Menick, Pablo Samuel Castro, Erich Elsen, ICML 2020
>
> [2] "The Elastic Lottery Ticket Hypothesis", Xiaohan Chen, Yu Cheng, Shuohang Wang, Zhe Gan, Jingjing Liu, Zhangyang Wang, NeurIPS 2021
>
> [3] "Drawing Early-Bird Tickets: Towards More Efficient Training of Deep Networks", Haoran You, Chaojian Li, Pengfei Xu, Yonggan Fu, Yue Wang, Richard G. Baraniuk, Yingyan Lin, Xiaohan Chen, Zhangyang Wang, ICLR 2020

---

### Official Review · Reviewer_W7x5 · 2021-10-30

**Correctness:** 3
**Technical Novelty And Significance:** 3
**Empirical Novelty And Significance:** 3
**Recommendation:** 8
**Confidence:** 4

**Main Review:**

Pros:

- The method is well-motivated. The sparse training techniques are now more and more important as the model sizes are increasing.
- The paper is overall clearly written, and the concept is easy to understand.
- The experiment results show the superiority of the proposed method compared to other methods like vanilla LTH.

Cons:

- I am not sure if this work can be seen as a complement/supplement/extension to the LTH. One key point in LTH is "trained in isolation", which means that the sparse sub-networks should be extracted out. In this work, however, the subnetwork emerges late, and all the connections in models are alive during the training. Also, there is also no such phase as re-training. Based on the comments above, I think this paper is more like a pruning paper aiming at compressing models and training models at the same time.
- If the above comments are correct, then I think the authors should provide a comparison between existing compressing methods. More specifically, pruning-after-training methods seem to be the correct way to look at.
- Some notions should be explicitly defined and some need correction. For example, the \bar{\theta^*} was never pre-defined. Also, the second condition in Equation (4) may never be satisfied. Changing "=" to be "\lt" would be better.

Minor: Some citation formats should have parentheses. Consider using \citep.





**Summary Of The Paper:**

The paper proposes a new pruning method that resembles distillation - keeping pruned connections within the pruned network and using them when generating the models' output. The method works better than the currently popular topic - LTH, and works better than some pruning-at-initialization methods such as GraSP.

**Summary Of The Review:**

My final score will be mostly based upon discussion with authors regarding my points outlined above. Currently, I think the paper is somehow off-topic, but I am open to hearing from the authors and willing to change my view if I had made mistakes.

---

> ### Author Response · Authors · 2021-11-18
> **Response To Reviewer W7x5**
>
> We appreciate Reviewer's insightful comments and we respond as below.
>
> We first clarify the relationships between "LTH" and "DLTH":
>
> **(1)** Both LTH and DLTH are articulated to study sparse network training -- trying a find a way to overcome the difficulty of training sparse networks directly. In addition, for both LTH and DLTH, we conduct the finetuning process with the same settings and hyperparameters to evaluate the sparse network training performance.
>
> **(2)** Based on the same goal -- finding a trainable sparse network, the duality between LTH and DLTH can be described in two ways:
>
> **(i)** LTH claims that the trainable subnetwork exists in a random dense network and finds it by pretraining and magnitude pruning. DLTH (ours) claims that each random subnetwork is trainable after a transformation and the transformation method is our proposed RST. (Please note, subnetwork picked randomly from a random dense network is hard to be trainable directly without any changes. This is shown in the comparison between "Scratch" with "RST" in our Table2.)
>
> We abbreviate this working pipeline duality as below:
>
> LTH: trainable subnetwork exists but unknown $-->$ pre-train + pruning $-->$ find it.
>
> DLTH: subnetwork can be freely assigned (we know its structure) but not trainable $-->$ transform $-->$ make it trainable
>
> **(ii)** Sparse network has two characteristics: network structure and network weights. As long as a sparse network is fixed, its structure and weights are fixed simultaneously, and they have to match with each other. Based on this, we describe the duality between LTH and DLTH as below:
>
> LTH: find structure according to weights (pruning using weight magnitude ranking).
>
> DLTH: find weights based on a given structure (transforming weights for randomly selected sparse network).
>
> Now, we make a response to the Reviewer's concerns:
>
> **(a) "trained in isolation" v. "emerges later":** Yes, the Reviewer's understanding is correct. LTH finds the existing trainable subnetwork. After finding it, it can be trained in isolation. DLTH transforms a randomly selected ticket into a trainable one. Thus, after the subnetwork is randomly selected and fixed, it cannot be trained in isolation and the winning ticket will emerge through the weights transformation (see (2).i above). This difference is also the entry point for us to articulate our work as a "dual problem" to the original LTH. If we conduct our work in a "trained in isolation" scenario, it is hard to work well, because a randomly selected ticket is very hard to be a winning ticket directly (see (2).i above).
>
> Let us further clarify it, from the research setting perspective, our DLTH formulates a working pipeline from a dual aspect of LTH to study sparse network training (see (2).i above), instead of following exactly the same scenario of LTH and making improvements within its studying framework. In addition, from a technical perspective, we formulate our DLTH as a dual problem of LTH as mentioned in (2).ii above.
>
> **(b) DLTH v. pruning:** Pruning methods have two settings: conventional pruning (pretrain+pruning) and pruning at initialization (random dense network + pruning). The key factor for both of them is using an effective criterion to remove redundant weights, like LTH to decide structure according to weight (see (2).ii above). On the contrary, DLTH allows us to decide the structure first and transform the weights based on the given structure. In other words, the random subnetwork selection in DLTH is very different from the operation of removing weight in conventional pruning based on its ultimate goal. Actually, from the angle mentioned in (2).ii above, conventional pruning and DLTH also have a duality relationship. Thus, although both DLTH and pruning are related to sparse network training, they study it from different aspects and based on different settings.
>
> We agree with the Reviewer's suggestion that the comparison with pruning methods is necessary. For conventional pruning, because it uses a pre-trained model, it is widely deemed to obtain higher performance [1]. We have already added the comparison with L1 pruning in Table2. For pruning at initialization, it selects a sparse network from a random dense network which is the same as DLTH. Thus, it is necessary to compare with it to show the superiority of DLTH. We compared the representative pruning at initialization method, GraSP, with our DLTH in Table3 and Table5
>
> Hope our explanation helps for a better understanding.
>
> For the Reviewer's other comments about notation and formats, we will carefully polish our draft for a better version.
>
> [1] "Pruning neural networks at initialization: Why are we missing the mark?", Frankle, Jonathan and Dziugaite, Gintare Karolina and Roy, Daniel M and Carbin, Michael, ICLR 2021

---

> > ### Comment · Reviewer_W7x5 · 2021-11-21
> > **Thank you for the response; score has been adjusted accordingly**
> >
> > After reading the authors' responses, I now have a better picture of the authors' work. They acknowledge that the setting is slightly different than the original LTH; however, it is indeed not necessary to strictly follow the LTH's setting as the authors argued - which I agree with. They have also provided new results (comparison with other pruning methods), showing their efforts during the rebuttal period. I have adjusted my score accordingly, but I have to say that I was expecting something more closer to the LTH setting. Hope the authors can work on it in future works.

---

> > > ### Author Response · Authors · 2021-11-23
> > > **Response To Reviewer W7x5**
> > >
> > > We thank the Reviewer's feedback and recognition of our rebuttal. In addition, we agree with the Reviewer's point that it is necessary and valuable to explore the sparse network training problem based on the setting more closer to the LTH. We will keep investigating this direction in our future works.
> > >
> > > Best wishes,
> > > Paper181 Authors

---

> > > > ### Comment · Reviewer_W7x5 · 2021-11-29
> > > > **Final score**
> > > >
> > > > After I have read other reviewers' comments and authors' responses, I feel more confident about the authors' work so I decide to give authors my support.

---

> > > > > ### Author Response · Authors · 2021-11-29
> > > > > **Response To Reviewer W7x5**
> > > > >
> > > > > We thank the Reviewer's continued attention to our work and we really appreciate the Reviewer's further support to our submission.
> > > > >
> > > > > Best wishes,
> > > > >
> > > > > Authors of Paper181

---

### Official Review · Reviewer_vhgZ · 2021-11-01

**Correctness:** 2
**Technical Novelty And Significance:** 3
**Empirical Novelty And Significance:** 3
**Recommendation:** 6
**Confidence:** 4

**Main Review:**

Major Pros:
That randomly selected sub-networks, can attain same performance as dense and comparable sparsity IMP derived LT has reached significant attention. The main idea presented here at tackling how a random network may be 'transformed' via RST is interesting.


Major Cons:
1. The main claim: " *any* randomly picked subnetwork .... " is incorrect. Examples where such a claim will fail are easy to construct: a choice of subnetwork which is disconnected, or lost its residual connection(s) or is woefully bottlenecked ...

2. The results with respect to cycle dependency are very noisy raising suspicion with respect to their statistical significance and/or the representativeness of the regime in which they were conducted. Why are they noisy (e.g. figure 5).
Specifically, for high sparsity levels --- performance in iterative magnitude pruning depends strongly on number of iterations (I assume this is what is meant by cycles?).

Style and clarity (not affecting score):
Throughout the paper much is left to the reader to infer burdening the clarity of the paper and its decipherability:
- At the onset of the main claim: what is a " trainable condition "
- Information Extrusion mentioned but not defined (as early as the abstract), only to be exemplified via implementation in section 5.
- 'Cycles' are not defined
- "As mentioned in implementation section, the extra training cost can be calculated..." the reader is referred to an equation in the Discussion of DLTH v. LTH section of the optimization cost , but no such equation is to be found in that reference.



*Updated review in light of author responses [Nov 26, 2021]:*

I have reviewed the updated manuscript following several iterations with the authors:  https://anonymous.4open.science/r/DLTH_Updated-3020/2022_ICLR_DLTH_Updated.pdf.

The updated manuscript changes the erroneous original claims ("any network can be transformed to trainable condition") and precisely modifies and scopes them ("random ticket wins"). In particular, giving room for a key novelty in the paper in the form of RST.

Issues with respect to the regime in which the conclusions are made remain, including an insufficient analysis of the sensitivity to iteration and duration of extrusion. These issues *should be expanded further in the final manuscript by additional experiments* in the regime where sensitivity is more pronounced (e.g. reaching higher sparsity levels where error materially increases --- and high sensitivity to pruning details is typically exhibited). Some concerns with respect to the generalizability of these results also remain.

However, with the updated discussions and revisions as well as the correction of the core claims, the paper rises to the level of acceptance --- assuming addressing the above issues, which the authors demonstrate willingness to do.
I thus accordingly update my score and thank the authors for their efforts in addressing concerns and for the open and constructive discussion and modifications.


**Summary Of The Paper:**

The paper claims that "Any randomly picked subnetwork in a dense randomly initialized neural network can be transformed into a trainable condition, where the subnetwork can be trained in isolation and achieves better at least comparable performance to LTH and other strong baselines."

In order to demonstrate how such a transformation may be achieved the paper introduces RST which gradually penalizes the magnitude (L2) of the non-selected parameters during training (upon which they are zeroed out explicitly) in order to arrive at the randomly selected sparse sub-network.



**Summary Of The Review:**

The paper's main claim as presented seems to be false --- it is expected not to hold true that *any* randomly picked subnetwork is transformable to a network as performant as the original dense network if trained.
For the particular method presented (RST) dependency on cycles leaves utility / insight unclear.

*Updated review in light of author responses [Nov 26, 2021]:* see above update. claims corrected, RST novel contribution, experiments for high sparsity in order to address high-sensitivity regime expected.

---

> ### Author Response · Authors · 2021-11-22
> **Response To Reviewer vhgZ**
>
> We thank for the Reviewer pointing out the drawback of our draft and we make the response as below.
>
> **DLTH description:** We appreciate the Reviewer's meticulously reading. In our DLTH, we mainly consider the common cases for randomly selected subnetworks from the dense model. Specifically, we conduct our experiments based on the uniformly random selection for weights with consistent layer-wise pruning ratios. We also evaluate the effectiveness of our RST based on different settings of layer-wise sparsity in the comparisons with GraSP. According to these experiments, our DLTH can be generally validated for common cases, however, it does not consider the extreme situations such as the disconnected subnetworks. We will revise the description of our DLTH and further clarify this point in our final version to eliminate any potential misunderstanding.
>
> **Experimental regime of the number of cycles:** In order to make fair and persuasive experiments in our work, we first conduct an ablation study on the LTH baseline to choose the most competitive setting for it. Based on ablation results shown in Figure3, we find "5" is the best number of iterative cycles for LTH, thus, we choose "5" as the number of iterative cycles for both LTH and DLTH, which is named as "LTH Iter-5" and "RST Iter-5" in our Table2. In this way, we tune the LTH baseline as strong as we can for comparison and solidly validate the effectiveness of our RST and DLTH.
>
> **Cycle number ablation study for our RST:** As mentioned in our response to the previous point above, we choose "5" as cycle number to conduct our main comparison experiments in Table2. After that, we further analyze our RST based on different cycle numbers, and the results are shown in Figure5. According to the visualization results, we find that, in most cases, using iterative strategies (cycle number: 2, 3, 4, 5) obtains better performance than the one-shot strategy (cycle number: 1). However, we also observe that the patterns of correlations between the number of cycles and the final performances for iterative strategies (cycle number: 2, 3, 4, 5) are not very obvious. The potential reason is adding relatively small iteration cycles is enough to enhance the capacity of RST, thus, it shows the robustness on different iteration numbers. We will keep investigating this problem in the future to achieve further performance improvement.
>
> Please note, the "cycles" means the number of iteration. we will clarify this point in our final version.
>
> **Trainable condition:** The original LTH claims the subnetwork found by iterative magnitude pruning (IMP) has better trainability. In our work, we start from a complementary direction of LTH for sparse network training. Thus, we regard the subnetwork obtained from LTH as the trainable subnetwork (or the subnetwork in trainable condition) and set its performance as our benchmark to validate the effectiveness of our RST and the proposed DLTH. We will clarify this point in our final version.
>
> **The equation of the optimization cost:** The equation of calculating the optimization cost just follows the sentence mentioned by the Reviewer. The reason we refer to the "implementation section" is the specific numbers of the variables in the following equation are provided in the "implementation section". We use the numbers mentioned in that section to obtain the specific optimization costs as described in the rest of that paragraph. We will polish these descriptions to eliminate any confusion.
>
> Hope our explanation can help for a better understanding of the motivation and logic of our work and clear the Reviewer's concerns.
>
> For the Reviewer's other comments about our writing style and unclear description, we will supplement the necessary content and polish our draft to deliver a better version.

---

> > ### Comment · Reviewer_vhgZ · 2021-11-22
> > **Reply**
> >
> > I thank the authors for their detailed response including clear clarifications and for the revisions promised to address the concerns raised.
> >
> > The RST method is indeed interesting and holds promise for insights.
> >
> > However, the poor correlation of the cycle dependency (e.g. noisy figure5 ) origin remains unclear. This point, as mentioned, calls into question potentially the regime and thus conclusions validity. Under an optimistic view, the authors claim, this is a sign of "robustness", but this is not convincingly supported by the presented result.
> >
> > At its current state the paper holds promise but requires considerable revision and some further experimental investigation --- I believe that after such improvements it can make a potentially much stronger paper as it's core (revised / re-scoped) RST contribution is very interesting.
> > I thus leave my score unchanged as I can not evaluate, at the paper's current revision, such material modifications, but strongly encourage the authors to re-submit following said modifications [or offer a paper revision such that the modifications may be evaluated within the rebuttal window].

---

> > > ### Author Response · Authors · 2021-11-23
> > > **Discussion To Reviewer vhgZ**
> > >
> > > We appreciate the Reviewer's feedback. We make further explanations to clarify the Reviewer's concerns based on several key points as below.
> > >
> > > **Correlation is unclear:** As mentioned by the reviewer, the patterns in Figure5 are noisy and the correlation is unclear. We presume the Reviewer expects a positive correlation between iteration numbers and final performance (namely, higher iteration number has higher accuracy), just like what LTH performs (the blue line in Figure3), but for our DLTH, there is no clear correlation. Actually, this pattern is still aligned with our expectations and supports our conclusions. Let us explain it from the implementation details.
> > >
> > > For both LTH and DLTH based on iteration strategy with a given pruning ratio, the reserved subnetwork is selected or pruned iteratively and reaches the given sparsity finally. However, the iterative operations for LTH and DLTH are different. For each iteration of LTH, it first obtains the pre-trained model, then uses L1 pruning to remove the redundant weights based on the weight magnitude ranking. In this way, the subnetwork is elaborately selected in each iteration and LTH is progressively enhanced to achieve higher performances as shown in the blue line in Figure3. On the other hand, in each iteration, DLTH randomly selected the weights to add regularization and remove them after this iteration. This iteration strategy does improve the performances for most cases like the comparison between RST and RST Iter-5 in Table2 as the extra weights are gradually removed through each iteration, which makes the whole process more stable. However, compared with LTH enhanced by iteratively employing L1 pruning, this progressive enhancement process is relatively weak if we add more iterations for DLTH. Hence, there are no clear correlations between the cycle numbers and the final performance and this observation supports the robustness of our current RST model with iterative strategy.
> > >
> > > For how to wisely adjust the strategy of RST for a better transformation and RST with the iterative strategy for a further model enhancement, as we mentioned in the last response to Reviewer vhgZ and the discussions with Reviewer mwor, we will keep investigating this point in the future.
> > >
> > >
> > > **Y-axis scale in Figure5:** As visualized in Figure5, the lines appear noisy. The reason is the Y-axis scale in Figure5 is small, which makes the performance variation look large in this plot. If we observe the performance variations in Figure3, we will find that the DLTH actually has more *consistent* performances, especially compared with LTH which achieves a large improvement by adding more iterations, and our DLTH still generally outperforms LTH. We admit that this figure is not well-plotted. We will visualize it in an appropriate way to eliminate any confusion.
> > >
> > > **Robustness:** According to the discussions above, the consistent performances of DLTH based on different iteration numbers show the robustness of our RST with a little performance improvement. This is also one of the advantages of our model as it uses fewer iterations to achieve promising performances compared with LTH, which requires more iterations.
> > >
> > > For how to leverage more iterations to further improve performance, as we mentioned in the response to "Correlation is unclear" above, we leave it into our future work.
> > >
> > > Hope our further explanations can help for a clearer understanding of our paper.

---

> > > > ### Comment · Reviewer_vhgZ · 2021-11-23
> > > > **further discussion**
> > > >
> > > > I appreciate the further clarifications. They are consistent with my reading of the paper.
> > > > The iteration dependence is indeed important, as is the dependence on the tapering in the extrusion process (i.e. how long a training/transformation are the masked weights given).
> > > > The correlation/noise discussed is present also in figure 3 where scale of error is not zoomed in, presenting what looks more like a statistical variation over limited repetitions. The point is different: if the performance is already in an area where error is weekly dependent on iteration, it would be insightful to explore e.g. deeper compressions where error dependency is expected to be more sensitive to the problem constituents such as number of iterations and masked weights transformation duration; as well as pitfalls of poor random sub-networks.
> > > >
> > > > I am happy to promptly re-evaluate and am very open to increasing my score if I can evaluate an updated manuscript which incorporates the substantial degree of material revisions and discussions the authors have committed to [e.g. via anonymized link to pdf/arxiv ]. My view is that this paper holds significant promise as a core idea of RST, after agreed material corrections of the claims [uniform sampling (not *any* topology) if shown on diversity of networks and well vetted empirically] is significant --- and can be revised to a strong submission.

---

> > > > > ### Author Response · Authors · 2021-11-26
> > > > > **Discussion To Reviewer vhgZ**
> > > > >
> > > > > We thank the Reviewer's further feedback for discussion and we follow the Reviewer's suggestion to provide an updated manuscript.
> > > > >
> > > > > Please check this anonymized link for the updated PDF file: https://anonymous.4open.science/r/DLTH_Updated-3020/2022_ICLR_DLTH_Updated.pdf.
> > > > >
> > > > > We briefly summarize our updated manuscript as below:
> > > > >
> > > > > 1) Compared with our original submission, our updated version has the same contributions and conclusions. We update our draft without changing the core part of our paper. Also, we try to keep the original content as much as we can and make necessary revisions for a clear version.
> > > > >
> > > > > 2) For revision, we integrate all our discussions about the Reviewer's concerns in this new version and polish the whole draft to keep the corresponding descriptions consistent. We use the blue color in the draft to represent all revisions including adding discussions and revising inappropriate descriptions.
> > > > >
> > > > > 3) Particularly, for the DLTH cycle number ablation study, we retain Figure5 and further add Table4 to thoroughly discuss this point by adding the corresponding discussions in the "Cycle Number Ablation Study for DLTH" section.
> > > > >
> > > > > Hope our updated manuscript can help clear the Reviewer's concerns.

---

> > > > > > ### Comment · Reviewer_vhgZ · 2021-11-27
> > > > > > **Final reply**
> > > > > >
> > > > > > Please see my updates to the review --- and updated score above.

---

> > > > > > > ### Author Response · Authors · 2021-11-27
> > > > > > > **Response To Reviewer vhgZ**
> > > > > > >
> > > > > > > We thank the Reviewer's approval of our rebuttal and recognition of our work. We really appreciate the Reviewer's suggestions and discussions for us to improve our paper.
> > > > > > >
> > > > > > > Best wishes,
> > > > > > >
> > > > > > > Authors of Paper181

---

### Official Review · Reviewer_XCnY · 2021-11-09

**Correctness:** 2
**Technical Novelty And Significance:** 2
**Empirical Novelty And Significance:** 3
**Recommendation:** 6
**Confidence:** 5

**Details Of Ethics Concerns:**

No ethics concerns regarding this work.

**Main Review:**

Two main concerns I have on this work are the way it is posed and the writing issue.

For how this work is posed, as the authors mentioned in multiple places in the write-up, DLTH is about a proper "transformation" of any ticket in the lottery pool into the winning ticket. Correspondingly, the authors proposed the RST as the transformation above to substantiate DLTH. However, according to the formulation in eqn. (3) and (4), Random Sparse Network Training seems to a better name for RST as far as I can see because it looks more like a general (potentially a uniformly better one) training method for sparse neural networks from scratch. For example, we could for sure apply RST to the winning tickets found by IMP directly. Will combining IMP + RST give us a better performance than RST + any tickets? This experiment will give us a clue of which one of the following is more true (or both not true):

1) a winning ticket achieves the best performance regardless of the training method and any transformed ticket with RST can not achieve better; OR
2) the winning tickets are still better subnetworks and can also be improved in terms of accuracy with RST?

If the first case is more true, then the contribution of this work would be much more interesting to me. If the second is more true, however, I think this work is not posed well. It is not about transforming any ticket into a winning one, but about proposing a general better training method for sparse neural networks. Let alone that the core regularization technique in RST is not new.

Another concern of mine is the writing. Firstly, the introduction and discussion of literature before Section 4 seem to be too long to me. Secondly, Section 3 is so disconnected with Section 4 and Section 5. Even the notations are not consistent in Section 3 and 5. The authors made efforts to explicitly define the transformation of weight and mask but lacked discussion about them in Section 4 and 5, like how RST is related to them, making RST just a training technique with regularization, totally losing connection to Section 3. I think this issue is also related to my first concern about the posing of this work. My suggestion is to shorten the contents before Section 3 and add richer discussion in Section 3, 4 and 5 to make them more logically consistent and connected.

**Summary Of The Paper:**

This work proposes the so called Dual Lottery Ticket Hypothesis which claims that every ticket in the ticket pool, i.e., every random sparse subnetwork in a randomly initialized dense neural network, can be transformed into a winning ticket with admirable trainability. This "transformation" proposed in the notation of Random Sparse Network Transformation (RST), in practice, comes in with the form of a squared L2 norm regularization on the masked weights, making them still involved during training while extruding the information contained in them and transferring into the unmasked ones. The empirical experiments show good empirical performance of RST compared to the vanilla LTH, pruning-at-initialization methods and LTH variants such as EB-LTH.

**Summary Of The Review:**

My evaluation for this work is that it is below the acceptance threshold, but I tend to give a score 4 (but I don't have option). Major reason is the posing of the DLTH claim and the contribution. I also think the writing could be fundamentally improved. I still think the technique and the idea are interesting and encourage the authors to try the next venue with enough improvements.

---

> ### Author Response · Authors · 2021-11-20
> **Response To Reviewer XCnY**
>
> We appreciate Reviewer's careful comments to point out our unclear description and we make the response as below.
>
> We summarize the motivation and insights for both LTH and DLTH to respond to the Reviewer's concern about **how this work is posed**.
>
> **LTH:** LTH is motivated by a practical necessity: "if a network can be reduced in size, why do we
> not train this smaller architecture instead in the interest of making training more efficient as well?" [1]. However, there is a problem for the motivation above: "sparse architectures produced by pruning are difficult to train from the start" [1]. We rewrite them as *"we expect to train a network with sparse structure, however, we need to find the appropriate weights to match this sparse structure, because random weights initialization for this structure damages final performance."* **In one word, a promising sparse network requires a just-right "structure-weights" combination.**
>
> Based on the entry point above, [1] claims "dense, randomly-initialized, feed-forward networks contain subnetworks ("winning tickets") that - when trained in isolation - reach test accuracy comparable to the original network in a similar number of iterations".
> This means the appropriate "structure-weights" combination exists in dense randomly-initialized networks. Further, LTH finds this kind of combination by magnitude pruning. Specifically, it ranks the magnitude of the weights of a pre-trained model to illustrate the corresponding structure.
>
> To make LTH achievable on the ImageNet dataset, researchers also study to find subnetworks early in training rather than at initialization [2]. This further confirms that the key point of LTH-related research is finding the proper structure-weights combination for sparse network training. Similarly, [2] still decides the structure based on weights magnitude ranking which is the same as the original LTH.
>
> **DLTH:** With the same goal of LTH -- finding the just-right "structure-weights" combination, we propose our DLTH. DLTH goes from a reverse direction: fixing structure then transforming the weights to match. Moreover, instead of finding one specific combination, DLTH allows arbitrarily selecting structure then makes its weights matched. Following the logic above, we pose our work, Dual Lottery Ticket Hypothesis (DLTH), as a dual problem of original LTH.
>
> We follow Reviewer's constructive suggestion for supplementary experiments. We further conduct experiments on CIFAR10/CIFAR100 with pruning ratios 0.5, 0.7, 0.9, 0.95, and 0.98. Results are shown below:
>
> |cifar10 | pr0.5 | pr0.7 | pr0.9 | pr0.95 | pr0.98 |
> |----|----|----|----|----|----|
> |LTH Iter-5 | 92.68 (0.39) | 92.50 (0.15) | 90.24 (0.27) | 88.10 (0.36) | 83.91 (0.15) |
> |RST |92.34(0.12)| 92.27(0.24)| **90.41(0.05)** |88.24(0.08) |83.77(0.47)|
> |LTH Iter-5+RST |**93.25 (0.03)** | **92.60 (0.16)** | 90.37 (0.35) | **88.56 (0.11)** | **85.46 (0.30)**|
> -----------------------------------
> | cifar100 | pr0.5 | pr0.7 | pr0.9 | pr0.95 | pr0.98 |
> | ---- | ---- | ---- | ---- | ---- | ---- |
> |LTH Iter-5  |70.57(0.15) | 69.54(0.46) | 64.84(0.11) | 60.45(0.61) | 53.83(0.09)|
> |RST |71.13 (0.48) |69.85(0.23) |66.17(0.18)| 61.66(0.37)| 54.11(0.37)|
> |LTH Iter-5+RST |**71.87 (0.29)** | **70.13 (0.26)** | **66.19 (0.05)** | **62.46 (0.30)** | **56.28 (0.26)**|
> -----------------------------------
> "LTH Iter-5" and "RST" are from Table2. We use our one-shot RST on the mask obtained by "LTH Iter-5" and refer it as "LTH Iter-5+RST". All results are run three times and reported as mean with standard deviation. We find the "LTH Iter-5+RST" obtains further improvement compared with "LTH Iter-5" which means RST improves the winning ticket performance; and also outperforms "RST" which means winning ticket is better than random ticket while both using RST. RST helps for obtaining an even better "structure-weights" combination for the structure of LTH by transforming its weights. This is in the line with our expectations and we believe this can further support our conclusions.
>
> We hope our explanation helps for a better understanding and solve the Reviewer's concerns.
>
> For the Reviewer's comments about writing and paper organization, we will polish our draft based on the discussions above and reorganize it accordingly.
>
> [1] "The Lottery Ticket Hypothesis: Finding Sparse, Trainable Neural Networks", Jonathan Frankle, Michael Carbin, ICLR 2019
>
> [2] "Linear Mode Connectivity and the Lottery Ticket Hypothesis", Jonathan Frankle, Gintare Karolina Dziugaite, Daniel M. Roy, Michael Carbin, ICML 2020

---

> ### Author Response · Authors · 2021-11-29
> **Sincerely Expecting Further Discussions with Reviewer XCnY**
>
> Dear Reviewer XCnY,
>
> We greatly appreciate the reviewing process so far! Given the ICLR final discussion deadline (11/29) is approaching, we really hope to have a further discussion with Reviewer XCnY to see if our responses solve the Reviewer's concerns.
>
> Thank you very much!
>
> Best wishes,
>
> Authors of Paper181

---

> > ### Comment · Reviewer_XCnY · 2021-11-29
> > **Just figured out a major misunderstanding of my previous evaluation**
> >
> > Dear authors,
> >
> > After carefully reading the discussions between you and other reviewers and the revised manuscript, I just figured out that I might have a major misunderstanding in my previous review.
> >
> > Previously, I thought RST was just part of one run of normal training and hence I considered it as an advanced but general training method for sparse neural networks, instead of the so called dual lottery tickets. Now I realize that the RST described in Sec. 5 corresponds to the $F_w$ mapping introduced in Sec. 3, eqn. (2), and it performs the re-conditioning of the remaining weights in any tickets by extruding the information from the masked weights into the remaining ones. Another run of standard SGD training will follow after running RST to evaluate the effectiveness of the transformed tickets. In this logic flow, this paper makes much more sense to me.
> >
> > However, I still think the writing of this work is not satisfying. The disconnection between Sec. 5 and Sec. 3-4 caused my misunderstanding in the role of RST. It was only mentioned in one sentence under the formal formulation of DLTH in Sec. 4 that "*$F_w$ represents our proposed random sparse network transformation (RST) process which will be detailed introduced in Sec. 5.*" And it is not mentioned anywhere that a standard training will follow after RST. I strongly recommend the authors to rewrite Sec. 5 to make that connection to Sec 3 and 4 much much clearer. Therefore, I will only raise my score to 6.
> >
> > Despite the raise of my score to 6, I have a new side question. It seems that the experiments did not show the effectiveness of RST. If we use a certain number of normal SGD steps as $F_w$ in place of RST (i.e., removing the extrusion regularization term), the resulting method will be similar to the setting of delaying the pruning described in Sec. 6 of [1]. How will it compared with RST? This would be nitpicking because my response has been late, for which I am sorry, and it is observed in [1] that delaying pruning does not help with random pruning. It is just out of curiosity and I personally think it would be better to have that ablation. This question will not influence my score.
> >
> > Overall, I am satisfied with the rebuttal from the authors, which helped to correct my misunderstanding. But I strongly recommend the authors to improve writing so that the big picture of the proposed DLTH is clearer.

---

> > > ### Author Response · Authors · 2021-11-30
> > > **Response To Reviewer XCnY**
> > >
> > > We appreciate the Reviewer's recognition of our rebuttal and feedback for discussions. We make the responses about the key points as below.
> > >
> > > **Missing the description of finetuning:** Actually, we mention this point mainly in the "Experiments" section. In the first sentence in the "Implementation" section, we mention that all experiments follow the same fine-tuning schedule for fairness with other details provided in the following content. Also, in the "Discussion of DLTH v. LTH", we provide specific numbers to analyze the computational resources including the finetuning process. We appreciate the Reviewer pointing out our unclear descriptions. We make further revisions as shown in this anonymous link, https://drive.google.com/file/d/109JavP0aFtKpuiUhT1zZS8bhtpbT2BrK/view?usp=sharing, to address the problem of the "unclear finetuning description". Please note, we try our best to keep the original version as it is and only make necessary revisions to make our draft clear and keep it consistent. In addition, the revision of this version is represented as the "orange" color and it is based on the revisions we made for the Reviewer vhgZ, which is represented as the "blue" color.
> > >
> > > **Overall writing structure:** We start from LTH to propose our DLTH with RST for validation. Hence, we introduce the LTH first with a general definition of network pruning which can be adaptively utilized for introducing the Pruning at Initialization (PI) and proposing our DLTH. Then, we describe our DLTH which is the proposed hypothesis in our work, also, naturally leading to our core method, RST, to validate it. After that, we introduce the details of RST to logically close the loop of our paper. We hope our response and the revised version provided above can help clear the relationships of our Sec 3, 4, and 5, and for a better understanding of our draft.
> > >
> > >
> > > **Ablation study of delaying the pruning:** We appreciate the Reviewer's suggestions about this ablation study and we agree with the Reviewer that it is a valuable comparison method. Currently, we are revising our code to implement this setting for evaluation. Due to the time limitation of the rebuttal period, we cannot guarantee we will respond to the Reviewer's concerns on time, however, if we can obtain the results timely, we will update them with analysis in the rebuttal window.
> > >
> > > Hope our responses can help clear the Reviewer's concerns.
> > >
> > > In addition, since the rebuttal deadline is approaching, we may not have a chance to further discuss with the Reviewer. Please let us know your further suggestions or concerns and we can improve our work and draft accordingly.

---

> > > ### Author Response · Authors · 2021-11-30
> > > **Response To Reviewer XCnY**
> > >
> > > We follow the Reviewer's suggestions to make the experiments. We summarize this experiment by following points:
> > > 1. If we use the normal SGD instead of RST, the previous RST process will be the regular SGD training.
> > > 2. For simplicity, we consider the one-shot strategy instead of the iteration strategy here. Actually, because it is based on regular SGD training with randomly selecting the subnetwork and without adding the regularization term, the one-shot and iteration strategies are equivalent to each other.
> > > 3. The whole pipeline will be "regular training + finetuning" with the subnetwork mask randomly selected in advance. In this way, this setting can be seen as a "random pruning" scenario, which requires: "regular SGD training" + "randomly selecting subnetwork" + "finetuning".
> > > 4. To make a thorough comparison, we set the "regular SGD training" as a "regular pre-training" process to obtain the strong version for this setting. Because if we only remove the regularization term and keep other hyperparameters as the same as our RST, the configuration is not an appropriate one for regular SGD training.
> > > 5. In this way, we have a new strong baseline seen as a regular "random pruning" method, which is based on a pre-trained model but a randomly selected subnetwork instead of magnitude-based selected subnetwork like L1 pruning.
> > > 6. We make quick experiments on CIFAR10 using ResNet56 based on pruning ratios: 0.5, 0.7, 0.9, 0.95, and 0.98. To save time, we only run each experiment once. The results are shown below.
> > >
> > > |Method|pr0.5|pr0.7|pr0.9|pr0.95|pr0.98|
> > > |----|----|----|----|----|----|
> > > |L1|93.33| 92.83| 91.67 |90.13|84.78|
> > > |LTH Iter-5|92.68| 92.50| 90.24| 88.10| 83.91|
> > > |Scratch|92.49| 92.14| 89.89| 87.41| 82.71|
> > > |DLTH Iter-5|93.41| 92.67| 90.43| 88.40| 83.97|
> > > |Random pruning|92.45|91.29|88.49|85.96|79.16|
> > >
> > > The first four rows are from Table2 in our draft and the last row is new results. We find that although the "Random pruning" is based on the pre-trained model, the randomly selecting operation may remove important weights and damage the final performance. Even if the model is trained from scratch, its performances are still a little higher than "Random pruning". We also find that the performance drop increases when the pruning ratio increases, which is also a reasonable observation. Because when we randomly remove the weights from a pre-trained model using a large pruning ratio, it is more likely to remove important weights which damages the final performance.
> > >
> > > Hope our further responses can help clear the Reviewer's concerns.

---

> > > ### Author Response · Authors · 2021-12-01
> > > **Response To Reviewer XCnY**
> > >
> > > We further supplement the experiments for the "ablation of delaying the pruning" on the CIFAR100 dataset using ResNet56. The results are shown below.
> > >
> > > |Method|pr0.5|pr0.7|pr0.9|pr0.95|pr0.98|
> > > |----|----|----|----|----|----|
> > > |L1|71.96| 71.59| 68.29| 64.74| 50.04|
> > > |LTH Iter-5|70.57| 69.54| 64.84| 60.45| 53.83|
> > > |Scratch|70.96| 68.59| 64.62| 59.93| 50.80|
> > > |DLTH Iter-5|71.39| 70.48| 65.65| 61.71| 54.46|
> > > |Random pruning|70.19|68.57|62.83|57.33|46.15|
> > >
> > > The first four rows are from Table2 in our draft ("DLTH Iter-5" represents the "RST Iter-5" in our draft). The last row is the new results. Similar to the results based on CIFAR10 provided in our previous response, we find the "Random pruning" achieves relatively low performance, even lower than "Scratch". In addition, for "Random pruning", there is also a trend that the performance drop increases when the pruning ratio increases. We conclude that the results of this ablation study on CIFAR100 are consistent with those on CIFAR10. And by adding this ablation study, we further demonstrate the effectiveness of the proposed RST and validate our DLTH.
> > >
> > > Hope our further responses can help clear the Reviewer's concerns.

---

### Decision · Program_Chairs · 2022-01-20

**Decision:**

Accept (Poster)

**Comment:**

I recommend this paper for acceptance but I do so with significant reservations. Since this metareview will be public for all time, I direct this metareview to future readers of this paper so that they can weigh its merits and drawbacks in a clear-minded way.

This paper proposes a "dual lottery ticket hypothesis." For those unfamiliar, the original lottery ticket hypothesis (Frankle & Carbin, ICLR 2019) states approximately that any randomly initialized neural network contains a subnetwork that can be trained in isolation to full accuracy in the same number of steps as the original network. That is, $\forall$ neural networks, $\exists$ a subnetwork such that $Accuracy(Train($subnetwork$)) \geq Accuracy(Train($network$))$ for a standard, fixed training procedure $Train$. (For the sake of posterity, note that this claim was supported on small-scale neural networks but there is not evidence that it holds in general; only that it holds on the state of networks *early* in training. See *Linear Mode Connectivity and the Lottery Ticket Hypothesis* by Frankle et al. 2020.) To support this claim, Frankle & Carbin develop a procedure that finds such subnetworks, demonstrating that they exist in certain settings.

As far as I understand, the dual lottery ticket hypothesis states that, $\forall$ subnetworks of a neural network, $\exists$ a setting of the weights such that $Accuracy(Train($subnetwork$)) \geq Accuracy(Train($network$))$. Like the original lottery ticket paper, this paper shows that such subnetworks exist: it trains the subnetwork with an L2 penalty on all of the weights except those of the subnetwork, allowing them to gradually fade away and leaving a new setting of the weights for the subnetwork that then allows it to train in isolation to full accuracy (like those subnetworks found in the original lottery ticket hypothesis paper).

The reason that I have reservations about this approach is that the subnetwork found by the dual lottery ticket hypothesis procedure contains fully trained weights. This is novel but - to me - much less surprising and interesting: a randomly sparse subnetwork can be set with trained weights such that, after all of the other weights are fully pruned away, it can recover full accuracy. On the one hand, this is almost reminiscent of a standard pruning procedure where the network is both trained and pruned until a sparse subnetwork reaches full accuracy, with the dense network needed for much or all of training. On the other hand, the impressive part is that this can be done with a *randomly selected* sparse network rather than one chosen by a pruning heuristic. To me, that is the most interesting part of the paper. (And, for those readers wondering why specifically this paper is distinct from standard pruning, this is it.)

I wonder about the significance of this finding given that the subnetwork is set by training (not by random initialization or a tiny amount of training as in work on the lottery ticket hypothesis), but it's a novel idea and I think future scholars and future research should be the judge of that significance, not me or the reviewers. The novelty alone merits publication, and we will have to wait and see about the significance. Thus, I weigh in favor of acceptance, although with reservations.